# Visual Encoders for Data-Efficient Imitation Learning in Modern Video Games

## Abstract

Video games have served as useful benchmarks for the decision making community, but going beyond Atari games towards training agents in modern games has been prohibitively expensive for the vast majority of the research community. Recent progress in the research, development and open release of large vision models has the potential to amortize some of these costs across the community. However, it is currently unclear which of these models have learnt representations that retain information critical for sequential decision making. Towards enabling wider participation in the research of gameplaying agents in modern games, we present a systematic study of imitation learning with publicly available visual encoders compared to the typical, task-specific, end-to-end training approach in Minecraft, Minecraft Dungeons and Counter-Strike: Global Offensive.

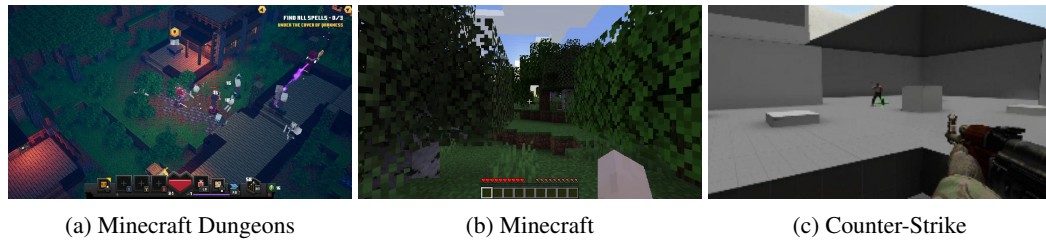

(a) Minecraft Dungeons         (b) Minecraft         (c) Counter-Strike

Figure 1: Representative screenshots of all games studied in this paper.

## 1 Introduction

Video games have served as useful benchmarks for the decision making community, training agents in complex games using reinforcement learning (RL) (Vinyals et al., 2019; Berner et al., 2019; Wurman et al., 2022), imitation learning (IL) (Kanervisto et al., 2020; Pearce & Zhu, 2022; Sestini et al., 2022), or a combination of both paradigms (Baker et al., 2022; Fan et al., 2022). However, video games do not only serve as benchmarks but also represent a vast entertainment industry where AI agents may eventually have applications in games development, including game testing or game design (Jacob et al., 2020; Gillberg et al., 2023).

In the past, video game research often necessitated close integration with the games themselves to obtain game-specific information and establish a scalable interface for training agents. Considering the costs associated with domain expertise and engineering efforts required for close integration, our focus is on training agents to play video games in a human-like manner, receiving only images from the game and producing actions corresponding to controller joystick and button inputs. To eliminate integration costs during training, we use behavior cloning to train agents entirely offline, utilising previously collected human gameplay data. Although prior research has explored encoding images into lower-dimensional representations for behavior cloning, these studies primarily targeted robotics applications (Nair et al., 2022), where images often resemble real-world scenes. Inspired by the challenges and potential applications in video games, we investigate the following research question: *How can images be encoded for data-efficient imitation learning in modern video games?*

Towards our guiding research question, we compare both end-to-end trained visual encoders and pre-trained visual encoders in three modern video games: Minecraft, Minecraft Dungeons and

Counter-Strike: Global Offensive (CS:GO). We examine 12 distinct end-to-end trained visual encoders, varying in network architecture (Residual Networks (ResNets) (He et al., 2016a;b) or Vision Transformers (ViTs) (Dosovitskiy et al., 2021; Steiner et al., 2022)), image input size, and the application of image augmentations. In contrast, pre-trained visual encoders are often trained on large datasets containing diverse real-world images, potentially providing useful and generalisable representations without additional training. However, it remains uncertain how well these pre-trained encoders perform in video games, which often exhibit substantial differences from real-world images. We identify four primary categories of training paradigms among pre-trained visual encoders and train agents using the representations from a total of 10 different pre-trained encoders spanning all these categories: (1) self-supervised trained encoders (e.g. DINOv2 (Oquab et al., 2023)), (2) language-contrastive trained encoders (e.g. OpenAI CLIP (Radford et al., 2021)), (3) supervised trained encoders (e.g. Imagenet classification trained FocalNet (Yang et al., 2022)), and (4) reconstruction trained encoders (e.g. the encoder of a variational autoencoder (VAE) (Kingma & Welling, 2013)). Finally, inspired by the cost associated with collecting human gameplay data and the hypothesis that pre-trained encoders could be advantageous in the absence of extensive training data, we investigate the performance of these visual encoders across varying amounts of training data.

Our results show that even though visual encoders trained end-to-end in complex video games can be effective with relatively small $128 \times 128$ images and limited amounts of high-quality data, substantial improvements can be achieved by employing pre-trained encoders, especially DINOv2.

## 2 RELATED WORK

**Learning Agents in Video Games.** Video games have often served as benchmarks for decision-making agents, leading to impressive performance in modern video games where a programmatic interface (Vinyals et al., 2019; Berner et al., 2019) or large quantities of expert demonstrations (Baker et al., 2022; Fan et al., 2022; Reed et al., 2022) are available for training. Recent work directly leverages or fine-tunes pre-trained foundation models to collect training data (Cai et al., 2023a) or guide action selection (Wang et al., 2023; Lifshitz et al., 2023; Cai et al., 2023b), but games without such close integration, extensive datasets or pre-trained models have seen comparably little research attention. Pearce & Zhu (2022) used imitation learning to train agents to play CS:GO with a combination of online scraped gameplay and expert demonstrations. Similarly, Kanervisto et al. (2020) benchmarked imitation learning agents across a diverse set of video games, including six modern games without programmatic interfaces. They emulated keyboard and mouse inputs to take actions in these games, akin to our approach. However, their study was limited to relatively simple visual encoders and agents did not leverage temporal history used in most recent decision-making agents.

**Visual Encoders for Imitation Learning.** Prior research has compared pre-trained visual encoders to those trained end-to-end using imitation learning for robotic applications (Nair et al., 2022; Yuan et al., 2022). These studies generally found that pre-trained encoders exhibit better generality and performance than those trained on smaller, task-specific data sets. However, given the real-world nature of robotics and the availability of datasets, it remains uncertain how these findings translate to the realm of video games. Our study seeks to bridge this gap.

**Visual Encoders for Video Games.** In the context of video games, pre-trained visual models have been employed to extract visual representations that differentiate between genres and styles (Trivedi et al., 2023), indicating their ability to detect relevant features in games. However, domain-specific models trained using self-supervised representation learning techniques can yield higher-quality representations than certain pre-trained visual encoders (Trivedi et al., 2022). Our study expands upon previous experiments by concentrating on modern video games and examining a broad spectrum of recent pre-trained and end-to-end trained visual encoder architectures.

## 3 IMITATION LEARNING FOR VIDEO GAMES FROM PIXELS

### 3.1 BEHAVIOUR CLONING

Behavior cloning (BC) is an imitation learning approach that trains agents through supervised learning using a dataset of provided demonstrations, denoted as $\mathcal{D} = (o_1, a_1), \ldots, (o_N, a_N)$, where $N$ represents the total number of samples in the dataset. Each demonstration comprises tuples $(o, a)$,

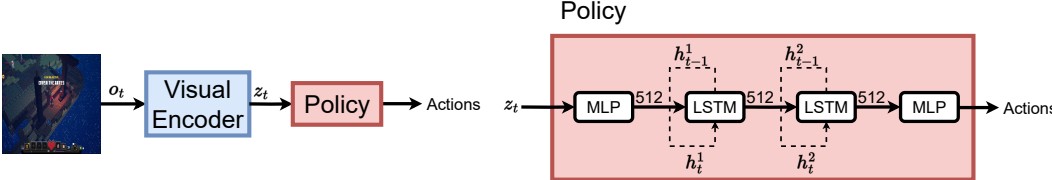

Figure 2: Architecture illustration: Core network architecture used throughout all the experiments.

which correspond to the image observation $o$ and the human player's chosen action $a$ at a specific point during training. Using this data, a policy $\pi(a \mid o; \theta)$ is trained to mimic the distribution of actions found in $\mathcal{D}$, based on the current image observed, by minimising the loss

$$\mathcal{L}(\theta) = \mathbb{E}_{(o,a)\sim\mathcal{D},\hat{a}\sim\pi(\cdot|o;\theta)} \left[l(a, \hat{a})\right] \tag{1}$$

where $l$ measures the discrepancy between the "true" action $a$ and the policy's sampled action $\hat{a}$. For continuous and discrete actions, we use the mean-squared error and cross-entropy loss, respectively.

## 3.2 IMAGE PROCESSING

Received images, sampled from the dataset during training or directly from the game during evaluation, are first resized to the required image size of the respective visual encoder (see Table 1)[1]. If image augmentation is used for an encoder, images are augmented after resizing using the same augmentations applied by Baker et al. (2022). Finally, image colour values are normalised.

## 3.3 ARCHITECTURE

The architecture of all trained agents is illustrated in Figure 2, and Table 1 lists all visual encoders considered in our experiments. The processed image $o_t$ is fed through the visual encoder to obtain an embedding $z_t$. The policy receives this embedding and outputs actions $a_t$ for the respective game.

**Policy Network.** For all experiments, the policy architecture remains identical. First, the received embedding $z_t$ is projected to 512 dimensions with a MLP with one hidden layer of dimension 512 before being fed through a two-layered LSTM (Hochreiter & Schmidhuber, 1997) with hidden dimensions of 512. The LSTM processes the projected embedding and a hidden state $h_{t-1}$ which represents the history of previously received embeddings during a sequence (obtained either as a sampled sequence during training or online evaluation). Following the two-layered LSTM, a MLP with one hidden layer of 512 dimensions is used to project the 512-dimensional representation outputted by the LSTM to as many dimensions as there are actions. At each intermediate layer, the ReLU activation function is applied.

**End-to-End Visual Encoders.** For visual encoders trained end-to-end with the BC loss, we consider three ResNet (He et al., 2016a;b) and three vision transformer (ViT) (Dosovitskiy et al., 2021; Steiner et al., 2022) architectures. The Impala (Espeholt et al., 2018) ResNet architecture is a commonly used visual encoder for decision making agents but designed for smaller image sizes than $128 \times 128$ and, thus, outputs large embeddings. For comparison, we evaluate two alternative larger ResNet architectures designed for images of size $128 \times 128$ and $256 \times 256$, respectively, which output smaller embeddings. For ViTs, we evaluate the commonly used tiny model architecture proposed by Steiner et al. (2022) which outputs fairly small embeddings. For comparison, we also evaluate two alternative architectures with slightly larger models that output comparably larger embeddings. See Appendix A.1 for full details on all end-to-end visual encoder architectures.

**Pre-Trained Visual Encoders.** We consider four paradigms of pre-trained visual encoders with representative models being evaluated in our experiments: OpenAI's CLIP (Radford et al., 2021) as language contrastive pre-trained encoders, DINOv2 (Oquab et al., 2023) as self-supervised pre-trained encoders with self-distillation objectives between a teacher and student network, FocalNet (Yang et al., 2022) trained on ImageNet21K classification as supervised pre-trained encoders, and a variational autoencoder (VAE) (Kingma & Welling, 2013) from stable diffusion (Rombach et al., 2022)

---

[1]For the resizing, we use linear interpolation for end-to-end encoders and bicubic interpolation for all pretrained encoders to be consistent with the processing pipeline used during training of the pre-trained encoders.

Table 1: Overview of all visual encoder architectures considered in this study including the type of training category, image sizes, parameter counts and the size of computed embeddings. For all encoders trained end-to-end with BC, we train them with and without image augmentation. For pre-trained models we only report the size of visual encoder used to embed images.

| Category | Model | Image size | Parameters | Embedding size |
|---|---|---|---|---|
| End-to-end | Impala ResNet | $128 \times 128$ | 98K | 7200 |
| | Custom ResNet | $128 \times 128$ | 585K | 1024 |
| | Custom ResNet | $256 \times 256$ | 586K | 1024 |
| | ViT Tiny | $224 \times 224$ | 5.5M | 192 |
| | Custom ViT | $128 \times 128$ | 8.8M | 512 |
| | Custom ViT | $256 \times 256$ | 8.9M | 512 |
| Language contrastive pre-trained | CLIP ResNet50 | $224 \times 224$ | 38M | 1024 |
| | CLIP ViT-B/16 | $224 \times 224$ | 86M | 512 |
| | CLIP ViT-L/14 | $224 \times 224$ | 303M | 768 |
| Self-supervised pre-trained | DINOv2 ViT-S/14 | $224 \times 224$ | 21M | 384 |
| | DINOv2 ViT-B/14 | $224 \times 224$ | 86M | 768 |
| | DINOv2 ViT-L/14 | $224 \times 224$ | 303M | 1024 |
| Classification supervised pre-trained | FocalNet Large FL4 | $224 \times 224$ | 205M | 1536 |
| | FocalNet XLarge FL4 | $224 \times 224$ | 364M | 2048 |
| | FocalNet Huge FL4 | $224 \times 224$ | 683M | 2816 |
| Reconstruction pre-trained | Stable Diffusion 2.1 VAE | $256 \times 256$ | 34M | 4096 |

as reconstruction pre-trained encoder. These visual encoders have already been trained on large amounts of real-world images. During all our experiments, we freeze these encoders and only use them to obtain embeddings of images without any fine-tuning or further training. For details on the models used, see Appendix A.2, and for details on the network architecture, training data, and further considerations of these encoders we refer to the original papers.

**Training Configuration.** For each network update, we sample 32 random sequences of 100 consecutive image-action pairs within the dataset. Before each training step, the hidden state and cell state of the LSTMs in the policy are reset and the mean BC loss is computed across all sequences with the hidden state accumulating across the 100 samples within a sequence. The Adam optimiser (Kingma & Ba, 2014) is used with decoupled weight decay (Loshchilov & Hutter, 2019) of 0.01 and a learning rate of $3 \cdot 10^{-4}$. To stabilise training, gradients are normalised at 1.0 and we use half precision for all training. In Minecraft Dungeons, we train each model for 1 million gradient updates. In Minecraft and CS:GO, models are trained for 500,000 gradient updates.

## 4 VIDEO GAMES FOR EVALUATION

We train and evaluate BC models with all visual encoders in three different games, Minecraft Dungeons, Minecraft and CS:GO, illustrated in Figure 1. Below, we will outline details regarding the training data and action space for each game.

### 4.1 MINECRAFT DUNGEONS

Minecraft Dungeons is an action-adventure role-playing video game with isometric camera view centered on the player. The player controls the movement and actions (including dodge roll, attack, use health potion, use items) of a single character which is kept in the center of the video frame (as seen in Figure 1a). The player has to complete diverse levels by following and completing several objectives. In our evaluation, we focus on the "Arch Haven" level of Minecraft Dungeons which contains fighting against several types of enemies and navigation across visually diverse terrain.

**Dataset.** Before data collection, we pre-registered this study with our Institutional Review Board (IRB) who advised on the drafting of our participant instructions to ensure informed consent. After

their approval, four players[2] played the "Arch Haven" level, and game frames at $1280 \times 720$ resolution, actions (joystick positions and button presses on a controller), and character position within the level were captured. The dataset includes a total of 139 recorded trajectories with more than eight hours of gameplay at 30Hz. Individual demonstrations vary between 160 and 380 seconds which corresponds to 4,800 and 11,400 recorded actions, respectively. We use 80% of the data for training and reserve 20% for validation. Each player was instructed to complete the level using a fixed character equipped with only the starting equipment of a sword and bow, and most players followed the immediate path towards level completion.

**Action space.** Agents have access to all effective controls in Minecraft Dungeons, including the x- and y-positions of both joysticks, the right trigger position (for shooting the bow), and ten buttons. The most frequently used buttons during recordings control sword attacks, bow shooting, healing potions, and forward dodging.

**Online evaluation.** To evaluate the quality of trained BC policies, we rollout the policy in the game with actions being queried at 10Hz (see Appendix D for details). These actions are then taken in the game using Xbox controller emulation software. Each rollout spawns the agent in the beginning of the "Arch Haven" level and queries actions until five minutes passed (3,000 actions) or the agent dies four times resulting in the level being failed. We run 20 rollouts per trained agent and report the progression throughout the level (Appendix C).

## 4.2 MINECRAFT

Minecraft is a game that lets players create and explore a world made of breakable cubes. Players can gather resources, craft items and fight enemies in this open-world sandbox game. Minecraft is also a useful platform for AI research, where different learning algorithms can be tested and compared (Johnson et al., 2016). We use the MineRL (Guss et al., 2019; Baker et al., 2022) environment, which connects Minecraft with Python and allows us to control the agents and the environment. We use MineRL version 1.0.2, which has been used for large-scale imitation learning experiments before (Baker et al., 2022), and which offers simpler mouse and keyboard input than previous MineRL versions (Guss et al., 2019).

**Task and online evaluation.** To evaluate our BC models, we use the "Treechop" task; after spawning to a new, randomly generated world, the player has to chop a single log of a tree within 1 minute. This is the first step to craft many of the items in Minecraft, and has been previously used to benchmark reinforcement learning algorithms (Guss et al., 2019). See Figure 1b for a screenshot of the starting state. The agent observes the shown image pixels in first-person perspective, can move the player around and attack to chop trees. For reporting the performance of trained models, we rollout each model for 100 episodes with the same world seeds, and record the number of trees the player chopped. If the player chopped at least one tree within the first minute, the episode is counted as a success, otherwise it is counted as a failure (the timeout is set to 1 minute).

**Dataset.** We use the Minecraft dataset released with the OpenAI VPT model (Baker et al., 2022) to select demonstrations of tree chopping. We choose the 6.13 version of the dataset and filter it to 40 minutes of human demonstrations that start from a fresh world and chop a tree within 1 minute. We also remove any erroneous files that remain after the filtering. The demonstrations include the image pixels seen by the human player at $640 \times 360$ resolution and the keyboard and mouse state at the same time, recorded at 20Hz. We also run the models at 20Hz.

## 4.3 COUNTER-STRIKE: GLOBAL OFFENSIVE

CS:GO is a first-person shooter game designed for competitive, five versus five games. The core skill of the game is accurate aiming and handling the weapon recoil/sway as the weapon is fired. Previous work has used CS:GO as a benchmark to train and test behavioural cloning models (Pearce et al., 2023), with best models able to outperform easier bots (Pearce & Zhu, 2022). We incorporate experiments using CS:GO, as it offers visuals more similar to the real-world images that most pretrained visual encoders were trained on, in contrast to our primary evaluation in Minecraft Dungeons and Minecraft (see Figure 1c).

---

[2]120 recordings were collected by one player with the remaining 19 recordings being roughly evenly split across the other three players.

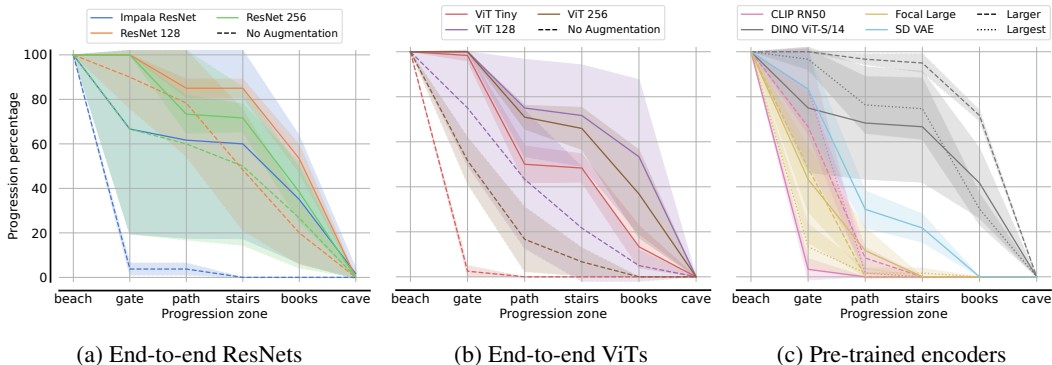

(a) End-to-end ResNets      (b) End-to-end ViTs      (c) Pre-trained encoders

Figure 3: Online evaluation progression for BC agents in Minecraft Dungeons with (a) end-to-end ResNets, (b) end-to-end ViTs, and (c) pre-trained visual encoders. We visualise the mean and standard deviation, computed over three training seeds, of the percentage of rollouts progressing through each of the objective zones within the "Arch Haven" level. Results for the ViT Tiny model are only aggregated over two seeds, as one seed resulted in an invalid checkpoint.

Following Pearce et al. (2023), we use the "Clean aim train" dataset and setup. The controlled player is placed in the middle of an arena, and random enemies are spawned around them who try to reach the player. The player can not move; they can only aim to different directions (Figure 1c). The dataset contains 45 minutes of expert-human gameplay from one player, recorded at 16Hz. To evaluate models, we run each model for three rollouts of five minutes each, and report the average and standard deviation of the kills-per-minute.

## 5 EVALUATION

In our evaluation, we focus on the guiding question of how to encode images for data-efficient imitation learning in modern video games. The evaluation is structured in four experiments studying (1) which end-to-end visual encoder is most effective, (2) which pre-trained encoder is most effective, (3) how do the best end-to-end and pre-trained visual encoders compare under varying amounts of training data, and (4) how do the best visual encoders compare in a video game with visuals more akin to the real-world. For each experiment, we train each model with three different seeds and report aggregated training and online evaluation metrics. Lastly, we visually inspect the visual encoders with respect to the information they attend to during action selection. An outline of the computational resources used for training and evaluation can be found in Appendix G.

### 5.1 HOW TO CHOOSE END-TO-END VISUAL ENCODERS?

To identify which end-to-end visual encoder is the most effective, we train all six end-to-end visual encoder architectures (listed in Table 1) with and without image augmentations using the BC loss. Figures 3a and 3b visualise the online evaluation performance for all models with end-to-end ResNet and ViT visual encoders, respectively, in Minecraft Dungeons. We observe that image augmentation consistently improves online performance, leading to models that more robustly progress further. ResNet image encoders slightly outperform ViT models, but by no significant margins. Lastly, no notable difference can be observed for end-to-end encoders trained on images of $128 \times 128$ or $256 \times 256$ resolution. These results suggest that comparably small images of $128 \times 128$ are sufficient even for complex video games like Minecraft Dungeons. In Minecraft (Table 2, left half), we also observe that the input image size has no significant effect on the results. However, ViT 256 and ViT Tiny outperform most ResNets by statistically significant margins (double-tailed Welch's test, $p < 0.05$) without image augmentations.

These results suggest two main findings: (1) Small images of $128 \times 128$ can be sufficient to train agents in complex modern video games, and (2) image augmentation has the potential to significantly improve performance but is game-specific.

Table 2: Minecraft online evaluation of agent success rate chopping a single tree with end-to-end trained (left) and pre-trained (right) visual encoders. Mean and one standard deviation computed over three training seeds. The best model in each group is highlighted in bold. Stars (*) indicate number of valid seeds averaged over if less than three, as some unstable runs resulted in invalid checkpoints.

| Model name | Success rate (%) |
|---|---|
| Impala ResNet[**] | $4.00 \pm 4.00$ |
| ResNet 128 | $12.67 \pm 3.86$ |
| ResNet 256 | $10.00 \pm 2.45$ |
| ViT Tiny | $23.33 \pm 4.19$ |
| ViT 128 | $19.00 \pm 2.94$ |
| **ViT 256** | $\mathbf{24.33 \pm 0.94}$ |
| Impala ResNet +Aug[*] | $14.00 \pm 0.00$ |
| ResNet 128 +Aug | $10.00 \pm 1.41$ |
| ResNet 256 +Aug | $6.67 \pm 1.70$ |
| ViT Tiny +Aug | $20.00 \pm 5.66$ |
| ViT 128 +Aug | $20.33 \pm 8.06$ |
| ViT 256 +Aug | $13.67 \pm 2.62$ |

| Model name | Success rate (%) |
|---|---|
| CLIP ResNet50 | $19.33 \pm 8.65$ |
| CLIP ViT-B/16 | $11.33 \pm 1.25$ |
| CLIP ViT-L/14 | $11.33 \pm 3.30$ |
| DINOv2 ViT-S/14 | $22.33 \pm 2.49$ |
| DINOv2 ViT-B/14 | $25.33 \pm 2.05$ |
| **DINOv2 ViT-L/14** | $\mathbf{32.00 \pm 1.63}$ |
| FocalNet Large | $16.00 \pm 5.66$ |
| FocalNet XLarge | $15.33 \pm 4.03$ |
| FocalNet Huge | $13.00 \pm 1.41$ |
| Stable Diffusion VAE | $20.00 \pm 5.89$ |

## 5.2 HOW TO CHOOSE PRE-TRAINED VISUAL ENCODERS?

To identify most suitable pre-trained visual encoders for video games, we compare BC agents trained with the representations of 10 pre-trained encoders. These encoders are frozen during BC training.

In Minecraft Dungeons (Figure 3c) and MineCraft (Table 2, right half), we find that BC models with DINOv2 visual encoders generally outperform other models. In Minecraft Dungeons, the BC models trained with DINOv2 ViT-B/14 pre-trained encoder outperforms all other models, including any end-to-end trained visual encoder. The stable diffusion encoder still outperforms FocalNet and CLIP visual encoders, but performs notably worse than all DINOv2 models. In Minecraft, the largest DINOv2 ViT-L/14, significantly ($p < 0.05$) outperforms all but the noisiest models (Tiny ViT, ViT 128 +Aug, Stable Diffusion and CLIP ResNet 50). While smaller DINOv2 models appear better than FocalNet or CLIP, their results are not significantly different from ViT-B/14 and ViT-S/14 DINOv2 models. Stable diffusion VAE works similarly to smaller DINOv2 models in Minecraft. Lastly, we observe that there is no clear correlation between the model size of pre-trained encoders and online performance. While larger DINOv2 models perform best in Minecraft, the same trend does not hold for CLIP and FocalNet where encoders with fewer parameters perform better.

## 5.3 HOW MUCH DATA DO YOU NEED?

A significant advantage of utilising pre-trained visual encoders is their independence from additional training, potentially resulting in more reliable performance with limited data. In contrast, visual encoders specifically trained for a particular task may be less generalisable but have the potential to outperform general-purpose pre-trained encoders. To test this hypothesis, we examine how the top-performing end-to-end and pre-trained visual encoders (based on online evaluation performance) compare with varying amounts of data.

In Minecraft Dungeons, we select the DINOv2 ViT-S/14, ViT-B/14 models, as well as the ResNet and ViT architectures on $128 \times 128$ images and image augmentation as the best-performing pre-trained and end-to-end trained encoders, respectively. We generate two reduced datasets with 50% ($\sim$ 4 hours) and 25% ($\sim$ 2 hours) of the training data by sampling trajectories uniformly at random. Each of the selected models is then trained on the 50% and 25% training datasets for 500 and 250 thousand gradient updates, respectively. Figure 4 shows the online evaluation performance of all models. As expected, we can see that the performance of all models gradually deteriorates as the training data is reduced. For pre-trained models, the larger DINOv2 ViT-B/14 outperforms the smaller ViT-S/14 when dealing with smaller datasets. Regarding end-to-end trained models, the ViT model's performance declines more rapidly with smaller data quantities compared to the ResNet. However, contrary to expectations, both end-to-end trained visual encoders yield performance comparable to pre-trained models in lower data regimes.

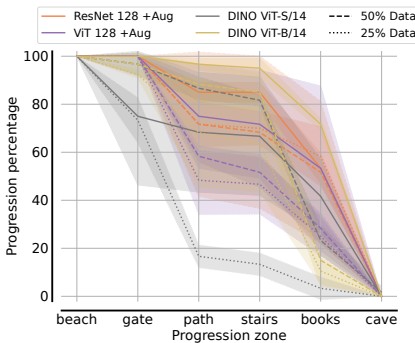

Figure 4: Online evaluation progression for the best-performing BC agents in Minecraft Dungeons with the full dataset (solid line) and subparts of the dataset.

Table 3: Online evaluation performance for the best-performing BC agents in Minecraft with the full dataset and 10% of the dataset.

| Model name | Success rate (%) |
|---|---|
| ViT 256 (Full) | $24.33 \pm 0.94$ |
| ViT 256 (10%) | $10.33 \pm 1.70$ |
| ViT Tiny (Full) | $23.33 \pm 4.19$ |
| ViT Tiny (10%) | $16.50 \pm 1.50$ |
| DINOv2 ViT-L/14 (Full) | $32.00 \pm 1.63$ |
| DINOv2 ViT-L/14 (10%) | $15.00 \pm 2.16$ |
| DINOv2 ViT-B/14 (Full) | $25.33 \pm 2.05$ |
| DINOv2 ViT-B/14 (10%) | $17.00 \pm 1.41$ |

In Minecraft, we also experimented with a low-data regime by using only 10% ($\sim$ 3.5 minutes, 14 demonstrations, 4269 steps of data) of gameplay data. The results for the two best end-to-end and pre-trained models are shown in Table 3. The success rate drops by half for all models, but it is still better than some models in the full experiments. This is surprising considering the small amount of data. Similar to Minecraft Dungeons, there is no significant difference between pre-trained and end-to-end visual encoders, suggesting that either of them could work well with less than 5 minutes of high-quality demonstration data. However, contrary to Minecraft Dungeons, there is no clear difference between both DINOv2 encoders in the lower data regime, suggesting that there is no clear correlation between model size and online performance in the very low data regime.

## 5.4 Grad-Cam Inspection of Visual Encoders

To understand what information is captured by visual encoders at various times in the games, we use gradient-weighted class activation mapping (Grad-CAM) (Selvaraju et al., 2017) to inspect each trained visual encoder. We visualise the Grad-CAM activations of visual encoders for images in Minecraft Dungeons and Minecraft with action logits of trained BC policies serving as the targets, these can be interpreted as which parts of the image are most relevant for the visual encoder during action selection. For more details on the Grad-CAM visualisations, plots for more game screenshots in both games and all visual encoders, see Appendix F.

Figure 5 shows the Grad-CAM activations for the best-performing visual encoders in both Minecraft Dungeons and Minecraft. In Minecraft Dungeons, many visual encoders tend to focus on the parts of the image containing the player character and enemy units. We hypothesise that other activations might correspond to way points the models focus on to navigate through the level. In Minecraft, most visual encoders tend to focus on parts indicative of nearby terrain, wood, and the progress of chopping a tree, aligning with the objective of the task.

## 5.5 Visual Encoders in CS:GO with More Realistic Visuals

To investigate visual encoders in a video game with more realistic images, akin to the training data of most pre-trained visual encoders, we evaluate ResNet 128 +Aug, ViT 128 +Aug and DINOv2 ViT-S/14 as the best-performing end-to-end and pre-trained visual encoders in CS:GO.

Table 4: Online evaluation performance in CS:GO as given by the kills-per-minute (KPM) in the aim training map. Mean and one standard deviation are provided.

| ResNet 128 +Aug | ViT 128 +Aug | DINOv2 ViT-S/14 |
|---|---|---|
| $7.97 \pm 0.57$ | $4.42 \pm 0.59$ | $2.18 \pm 1.12$ |

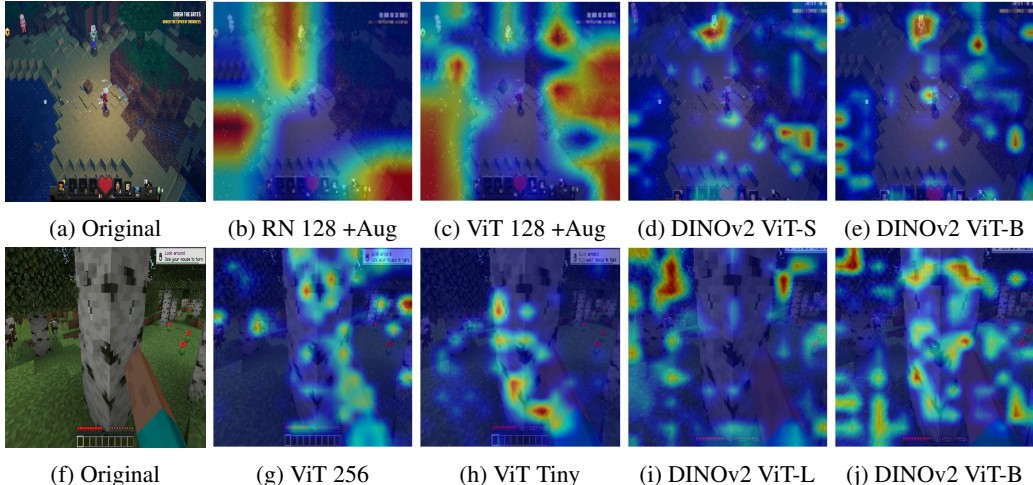

|           |           |           |             |             |
|-----------|-----------|-----------|-------------|-------------|
| (a) Original | (b) RN 128 +Aug | (c) ViT 128 +Aug | (d) DINOv2 ViT-S | (e) DINOv2 ViT-B |
| (f) Original | (g) ViT 256 | (h) ViT Tiny | (i) DINOv2 ViT-L | (j) DINOv2 ViT-B |

Figure 5: Grad-CAM visualisation of the activation of the best-performing visual encoders for Minecraft Dungeons (top) and Minecraft (bottom) with action logits of a BC policy serving as targets. Red areas represent the parts of the image the visual encoders focus on the most.

Results in Table 4 indicate end-to-end trained models perform significantly better ($p < 0.05$) than DINOv2, and ResNet outperforms ViT (also $p < 0.05$). Initially, we hypothesised that the image processing in CS:GO[3] might be the cause for the poor online performance of DINOv2. However, further investigation with four pre-trained visual encoders in Minecraft (detailed in Appendix E) indicates that pre-trained visual encoders are not as sensitive to the image processing as hypothesised from the performance of DINOv2 ViT-S/14 in CS:GO. Even with similar image processing as applied in CS:GO, agents trained with pre-trained visual encoders were able to exhibit performance comparable to our original findings in Minecraft. We leave further investigation into the efficacy of pre-trained visual encoders in CS:GO and the observed failure of DINOv2 for future work.

## 6 CONCLUSION

In this study, we systematically evaluated the effectiveness of imitation learning in modern video games by comparing the conventional end-to-end training of task-specific visual encoders with the use of publicly available pre-trained encoders. Our findings revealed that training visual encoders end-to-end on relatively small images can yield strong performance when using high-quality, representative data for the evaluation task, even in low-data regimes of few hours or minutes. DINOv2, trained with self-supervised objectives on diverse data, consistently outperformed other pre-trained visual encoders, indicating its generality and suitability for video games. Interestingly, agents using these pre-trained visual encoders demonstrated performance comparable (or superior) to those employing game-specific visual encoders across different data volumes. However, careful attention must be given to image resizing and processing as seen in CS:GO. Overall, our results suggest that the use of effective pre-trained visual encoders, such as DINOv2, should be seriously considered in the context of modern video games.

In order to maintain focus and feasibility in our study, we concentrated on a specific set of visual encoders, enabling a range of comparisons between different architectures and pre-trained model paradigms. Nevertheless, our study could be complemented by exploring additional comparison points, such as diverse supervised-trained pre-trained encoder architectures and additional scenarios within the examined or other video games. Although our study focused on settings with available training data for the evaluation task, future work could explore the potential benefits of pre-trained visual encoders when agents need to generalise across diverse levels or maps with variable visuals.

---

[3] The CS:GO dataset contains images at a resolution of $280 \times 150$ but images have to be resized to $224 \times 224$ for DINOv2. This up-scaling of the image height after initial down-scaling during the dataset collection differs from the image processing applied during the pre-training of DINOv2.

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

# A VISUAL ENCODERS

In this section, we will describe the architectures of all end-to-end visual encoders, the image augmentations applied for end-to-end visual encoders, and detail the sources of the pre-trained encoders used in our study.

## A.1 END-TO-END VISUAL ENCODERS

**Impala ResNet** The Impala ResNet architecture faithfully implements the visual encoder of the "large architecture" outlined by Espeholt et al. (2018) consisting of a $3 \times 3$ convolution with stride 1, max pooling with $3 \times 3$ kernels and stride 2 followed by two residual blocks of two $3 \times 3$ convolutions with stride 1. This joint block is repeated three times with 16, 32, and 32 channels, respectively.

**Custom ResNet** The architecture for our custom ResNet models is modelled after Liu et al. (2022) and illustrated in detail in Figure 6.

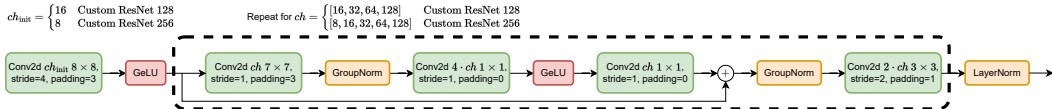

Figure 6: Illustration of the architecture of our custom ResNet visual encoders for $128 \times 128$ and $256 \times 256$ images.

**ViT** Our ViT architectures are all based on the reference implementation at `https://github.com/lucidrains/vit-pytorch/blob/main/vit_pytorch/vit.py`. For all models, we use no dropout, and the following configurations are used across the considered ViT visual encoders:

| Model name | Patch size | Num layers | Width | MLP dim | Num heads |
|---|---|---|---|---|---|
| ViT Tiny | 16 | 12 | 192 | 768 | 3 |
| Custom ViT | 16 | 4 | 512 | 512 | 12 |

Table 5: Configurations of end-to-end ViT models.

The ViT Tiny architecture follows the suggested architecture of Steiner et al. (2022). In contrast, both custom ViT for $128 \times 128$ and $256 \times 256$ have notably fewer layers, wider dimensions of the attention layers and no increase of dimensions in the MLP projections. In our experiments, we found that such an architecture resulted in better online evaluation performance in several video games.

**Image augmentations** If image augmentations are applied during training, we randomly augment images after the down-scaling process. We implement all augmentations with the `torchvision` library and randomly sample augmentations during training. We apply the following augmentations as described by Baker et al. (2022):

- Change colour hue by a random factor between -0.2 and 0.2
- Change colour saturation with a random factor between 0.8 and 1.2
- Change brightness with a random factor between 0.8 and 1.2
- Change colour contrast with a random factor between 0.8 and 1.2
- Randomly rotate the image between -2 and 2 degrees
- Scale the image with a random factor between 0.98 and 1.02 in each dimension
- Apply a random shear to the image between -2 and 2 degrees
- Randomly translate the image between -2 and 2 pixels in both the x- and y-direction

## A.2 Pre-Trained Visual Encoders

In this section, we will detail the sources for all pre-trained visual encoders considered in our evaluation.

**OpenAI CLIP**    For the visual encoders of OpenAI's CLIP models (Radford et al., 2021), we use the official interface available at `https://github.com/openai/CLIP`. We use the following models from this repository: "RN50" (ResNet 50), "ViT-B/16", and "ViT-L/14". In preliminary experiments, we found the available larger ResNet models to provide no significant improvements in online evaluation performance and the ViT model with a larger patch size of 32 to perform worse than the chosen ViT models with patch sizes of 16 and 14.

**DINOv2**    For the DINOv2 pre-trained visual encoders (Oquab et al., 2023), we use the official interface available at `https://github.com/facebookresearch/dinov2`. Due to the computational cost, we do not evaluate the non-distilled ViT-G/14 checkpoint with 1.1 billion parameters.

**FocalNet**    For the FocalNet pre-trained visual encoders (Yang et al., 2022), we used the Hugging Face *timm* library (`https://huggingface.co/docs/timm/index`) to load the pre-trained models for its ease of use. We use the FocalNet models pre-trained on ImageNet-22K classification with 4 focal layers: "focalnet_large_fl4", "focalnet_xlarge_fl4", and "focalnet_huge_fl4".

**Stable Diffusion**    For the pre-trained stable diffusion 2.1 VAE encoder, we use the Hugging Face checkpoint of the model available at `https://huggingface.co/stabilityai/sdxl-vae`. This model can be accessed with the *diffusers* library. In contrast to other encoders, the VAE outputs a Gaussian distribution of embeddings rather than an individual embedding for a given image. We use the mode of the distribution of a given image as its embedding since (1) we want to keep the embeddings of the frozen encoder for a given image deterministic, and (2) we find the standard deviation to be neglectable for most inputs.

## B  Additional Evaluation Data

In this section, we provide additional insight into our evaluation.

**Minecraft Dungeons**    Figure 7a (top) shows the training loss for all models with end-to-end visual encoders in Minecraft Dungeons. From the training loss, we can see that image augmentations generally increase the training loss despite improving online performance as seen in Figures 3a and 3b. We also note that training for the custom ResNet with $256 \times 256$ images and the Impala ResNet exhibit high dispersion across three seeds, leading to large shading and stagnation of the loss early in training. We hypothesise that this occurs for the Impala ResNet due to the overly large embeddings which complicate learning a policy with BC.

For BC models with pre-trained visual encoders, the training loss shown in Figure 7b appears comparably similar for most models. Only the reconstruction-based stable diffusion encoder and the CLIP ResNet50 models stand out since they outperform and underperform all other models, respectively.

Comparing the training loss of BC models trained with end-to-end and pre-trained visual encoders further shows that end-to-end encoders trained without image augmentation are capable of reaching lower losses. We hypothesise that this occurs since the end-to-end trained encoders are specialised to perform well on the exact training data the loss is computed over.

**Minecraft**    In contrast, the training loss in Minecraft (Figure 7 bottom) quickly stagnates and converges to similarly low values for all end-to-end and pre-trained encoders.

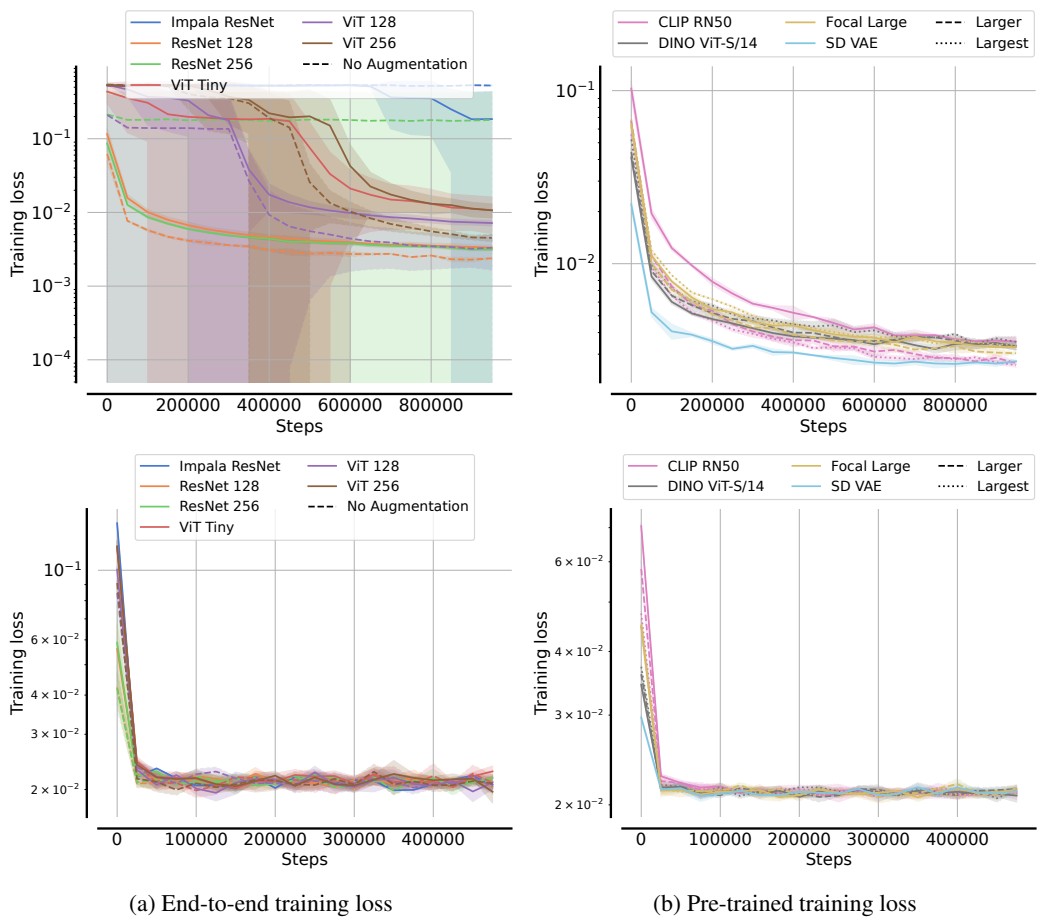

(a) End-to-end training loss

(b) Pre-trained training loss

Figure 7: Training loss in log-scale for BC agents in Minecraft Dungeons (top) and Minecraft (bottom) with (a) end-to-end trained and (b) pre-trained visual encoders. The training loss of all seeds at every step is averaged at twenty regular intervals throughout training. We visualise the mean and standard deviation across three seeds.

**CS:GO** In CS:GO, the training loss improves all throughout training for all three trained models with the models trained with DINOv2 ViT-S/14 pre-trained encoders achieving the lowest training loss. In contrast, both the end-to-end trained ResNet and ViT encoders trained with $128 \times 128$ images and image augmentation have higher training loss. We highlight that these end-to-end trained visual encoders are trained with image augmentations whereas the models with DINOv2 pre-trained encoders are not. Such image augmentations have been seen to consistently increase the training loss in Minecraft Dungeons and might be the main reason for the higher training loss of the end-to-end trained models.

# C MINECRAFT DUNGEONS ARCH HAVEN LEVEL

To measure progress for the online evaluation in Minecraft Dungeons, we define boundaries of zones which determine the progression throughout the "Arch Haven" level we evaluate in. These zones and a heatmap showing the visited locations of the human demonstrations used for training are visualised in Figure 9. The heatmap also shows the path followed by most demonstrations towards completion of the level.

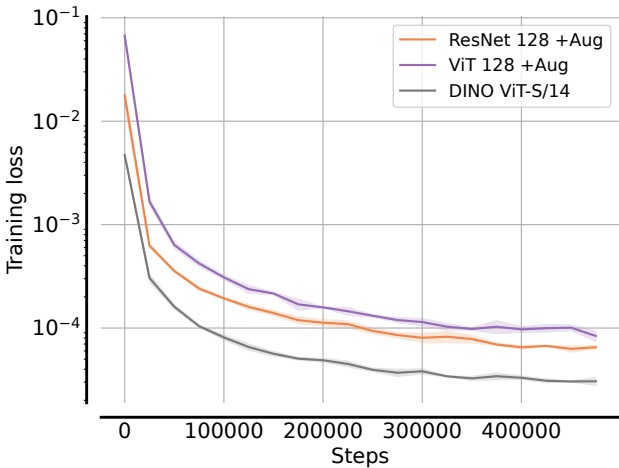

Figure 8: Training loss in log-scale for BC agents in CS:GO with (a) end-to-end trained and (b) pre-trained visual encoders.

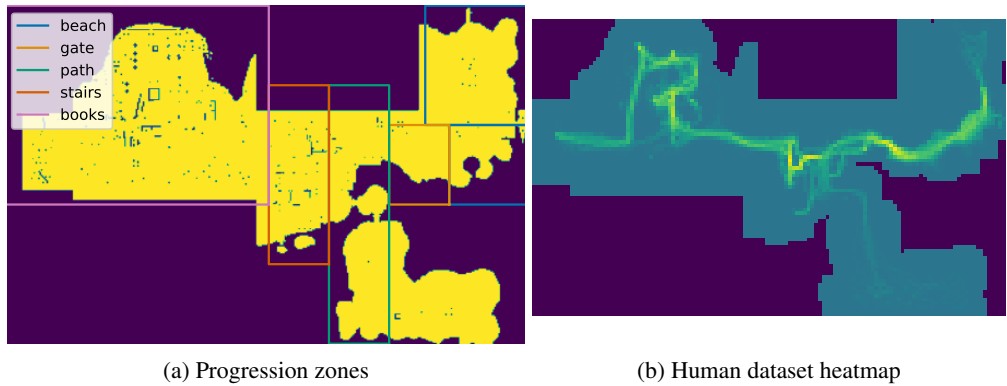

(a) Progression zones                    (b) Human dataset heatmap

Figure 9: (a) A visualisation of the boundaries of each progression zone in the "Arch Haven" level in Minecraft Dungeons used for online evaluations. (b) A heatmap visualising the visited locations of the human dataset of demonstrations within the "Arch Haven" level.

## D    MINECRAFT DUNGEONS ACTION FREQUENCY IN ONLINE EVALUATION

The visual encoders used in our evaluation have vastly different model sizes (see Table 1) and, thus, notably different computational cost at inference time. This is particularly challenging during online evaluation in Minecraft Dungeons, since there exists no programmatic interface to pause or slow down the game like in Minecraft and CS:GO. We attempt to take actions during evaluation at 10Hz to match the action selection frequency of the (processed) training data, in particular due to the recurrent architecture of our policy. However, we are unable to perfectly match this frequency for all visual encoders on the hardware used to conduct the evaluation (see Appendix G for specifications on the hardware used during training and online evaluation) despite using a more powerful GPU for pre-trained visual encoders due to their comparably large size.

Table 6 lists the average action frequencies of all models during online evaluation in Minecraft Dungeons across all runs conducted as part of our evaluation. We note that most end-to-end trained visual encoders enable fast inference achieving close to 10 Hz action frequency. The ViT Tiny model is the slowest model, likely due to its deeper 12 layers in comparison to the other end-to-end trained ViT models with 4 layers as shown in Table 5, but we are still able to take actions at more than 8.5Hz. For pre-trained visual encoders, we see comparably fast action frequencies for all CLIP and most DINOv2 models as. The largest DINOv2 and stable diffusion VAE have notably slower action frequencies, but the FocalNet models induced the highest inference cost. However, we highlight that

we did not observe behaviour during online evaluation which would suggest that these models were significantly inhibited due to this discrepancy.

Table 6: Average action frequencies during online evaluation in Minecraft Dungeons across 60 runs per model (20 for each seed).

| Model name | Action freq. (Hz) | Model name | Action freq. (Hz) |
|---|---|---|---|
| Impala ResNet | 9.83 | CLIP ResNet50 | 9.85 |
| ResNet 128 | 9.90 | CLIP ViT-B/16 | 9.84 |
| ResNet 256 | 9.81 | CLIP ViT-L/14 | 9.71 |
| ViT Tiny | 8.63 | DINOv2 ViT-S/14 | 9.81 |
| ViT 128 | 9.90 | DINOv2 ViT-B/14 | 9.81 |
| ViT 256 | 9.46 | DINOv2 ViT-L/14 | 7.93 |
| Impala ResNet +Aug | 9.78 | FocalNet Large | 8.00 |
| ResNet 128 +Aug | 9.67 | FocalNet XLarge | 6.13 |
| ResNet 256 +Aug | 9.62 | FocalNet Huge | 6.91 |
| ViT Tiny +Aug | 8.77 | Stable Diffusion VAE | 8.77 |
| ViT 128 +Aug | 9.69 | | |
| ViT 256 +Aug | 9.63 | | |

## E   IMAGE PROCESSING INVESTIGATION

As described in Section 5.5, we observe that a DINOv2 model performed poorly in CS:GO while this family of pre-trained visual encoders led to high online evaluation performance in both Minecraft Dungeons and Minecraft. We hypothesised that the cause of this poor performance in CS:GO is the image resizing. The CS:GO dataset includes images cropped and down-scaled to a resolution of $280 \times 150$ whereas DINOv2 pre-trained visual encoders (and most other pre-trained visual encoders) expect image sizes of $224 \times 224$. Therefore, images are down-scaled from a higher resolution to $280 \times 150$ during data collection and then up-scaled again to $224 \times 224$. We hypothesise that this resizing causes discrepancies in the resulting images compared to the expected processing of resizing images from a higher resolution directly to $224 \times 224$.

To verify this hypothesis, we conduct experiments in Minecraft instead of CS:GO since the dataset in CS:GO is only available with images of $280 \times 150$. In our original evaluation in Minecraft, we down-scaled images from a higher resolution of $640 \times 360$ to the respective resolution required by each visual encoder during training. To mimic the situation in CS:GO and separate the confounding factors of down-scaling followed by up-scaling and squared and non-squared aspect ratios, we consider two alternative image processing:

1. **CS:GO-like processing**: We down-scale images to width 280 (keeping aspect ratio), crop to $280 \times 150$ from the middle, and re-size from this resolution to the resolution required by the respective visual encoder.
2. **Squared aspect ratio**: We down-scale images from the dataset to $150 \times 150$ and re-size from this resolution to the resolution required by the respective visual encoder.

The first processing allows us to study how different visual encoders are affected by processing as done in CS:GO, whereas the second processing allows us to study how the same experiment behaves if squared aspect ratio is retained all throughout the process. We train and evaluate the best performing pre-trained visual encoder of each family of models (following Table 2) with the respective processing.

Table 7 shows the online evaluation performance of models trained with four different pre-trained encoders either using the original image processing, CS:GO-like processing, or the described squared processing. As we can see, the large DINOv2 as well as the CLIP ResNet encoder perform comparable for both the original processing and the CS:GO-like processing, but the performance of the

Table 7: Minecraft online evaluation of agent success rate chopping a single tree with varying visual encoders and image processing. Mean and one standard deviation computed over two seeds.

| Model name | Original processing | CS:GO-like processing | Squared processing |
|---|---|---|---|
| CLIP ResNet50 | $19.33 \pm 8.65$ | $19.00 \pm 6.00$ | $23.50 \pm 0.50$ |
| DINOv2 ViT-L/14 | $32.00 \pm 1.63$ | $32.00 \pm 4.00$ | $37.50 \pm 1.50$ |
| FocalNet Large | $16.00 \pm 5.66$ | $13.50 \pm 0.50$ | $17.50 \pm 0.50$ |
| Stable Diffusion VAE | $20.00 \pm 5.89$ | $15.50 \pm 0.50$ | $17.00 \pm 8.00$ |

FocalNet and Stable Diffusion encoders deteriorates slightly. Furthermore, all but the Stable Diffusion visual encoder perform best with the squared processing. Models trained with this processing, which ensures a squared aspect ratio for the initial down-scaling, perform best in terms of average performance and exhibit notably lower deviation across two runs. Overall, these results indicate that the processing applied in CS:GO is not necessarily detrimental to the performance of trained agents and maintaining a squared aspect ratio across image processing is desirable if resizing has to be done at several stages.

Given these results, the question as to why the smallest DINOv2 visual encoder performed poorly in CS:GO, as shown in Section 5.5, remains unanswered. We leave further investigation into the efficacy of pre-trained visual encoders for decision making in CS:GO for future work.

## F    GRAD-CAM VISUALISATIONS

To generate Grad-CAM (Selvaraju et al., 2017) visualisations, we use the library available at `https://github.com/jacobgil/pytorch-grad-cam`. We use all actions of the policy trained on the embeddings of each visual encoder as the target concept to analyse, and visualise the average Grad-CAM plot across all actions. Following `https://github.com/jacobgil/pytorch-grad-cam#chosing-the-target-layer`, we use the activations of these layers within the visual encoders to compute visualisations for:

- ResNet: Activations across the last ResNet block
- ViT: Activations across the layer normalisation before the last attention block
- FocalNet: Activations across the layer normalisation before the last focal modulation block
- SD VAE: Activations across the last ResNet block within the mid-block of the encoder

## F.1 MINECRAFT DUNGEONS

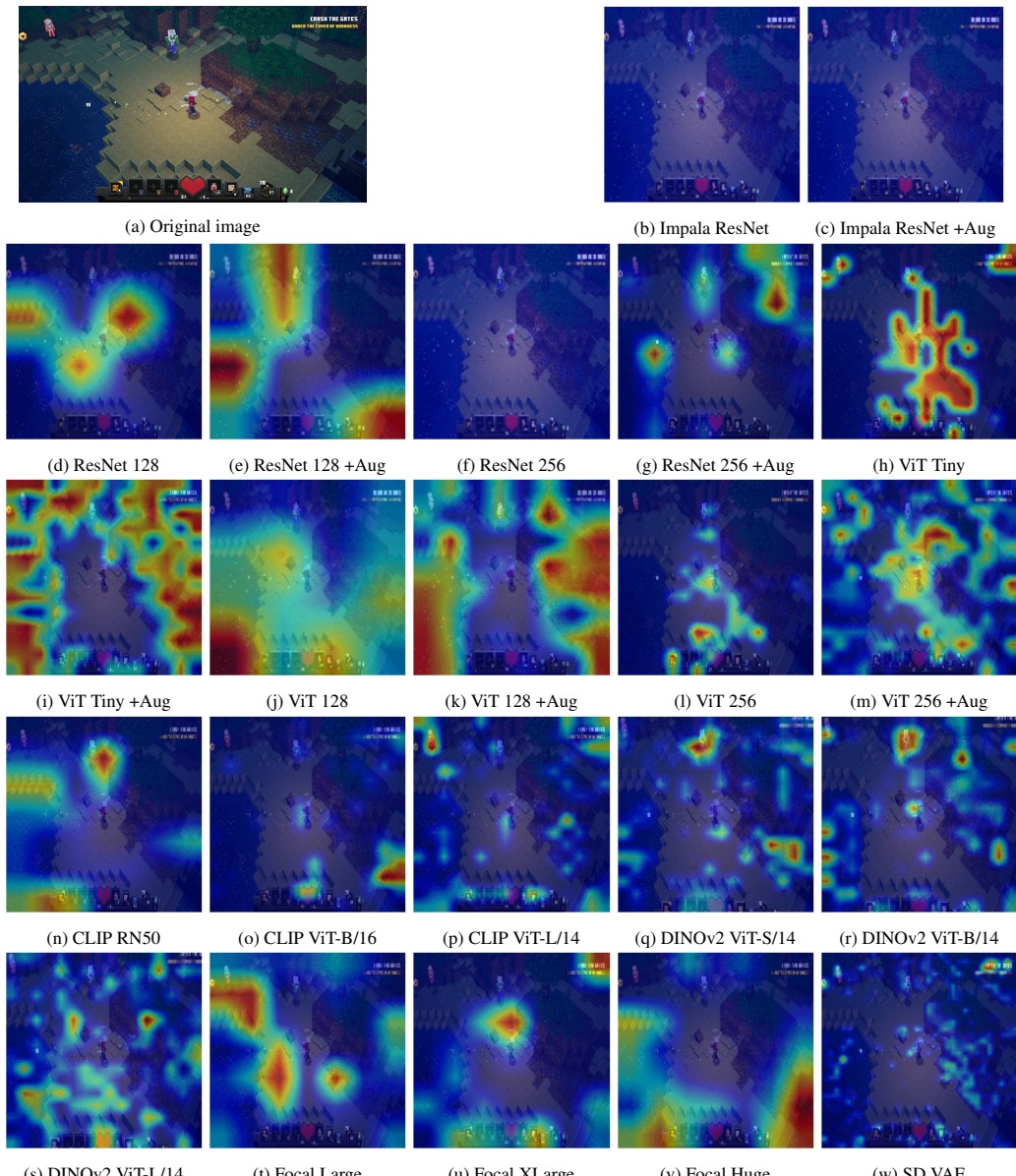

Figure 10: Grad-Cam visualisations for all encoders (seed 0) with policy action logits serving as the targets.

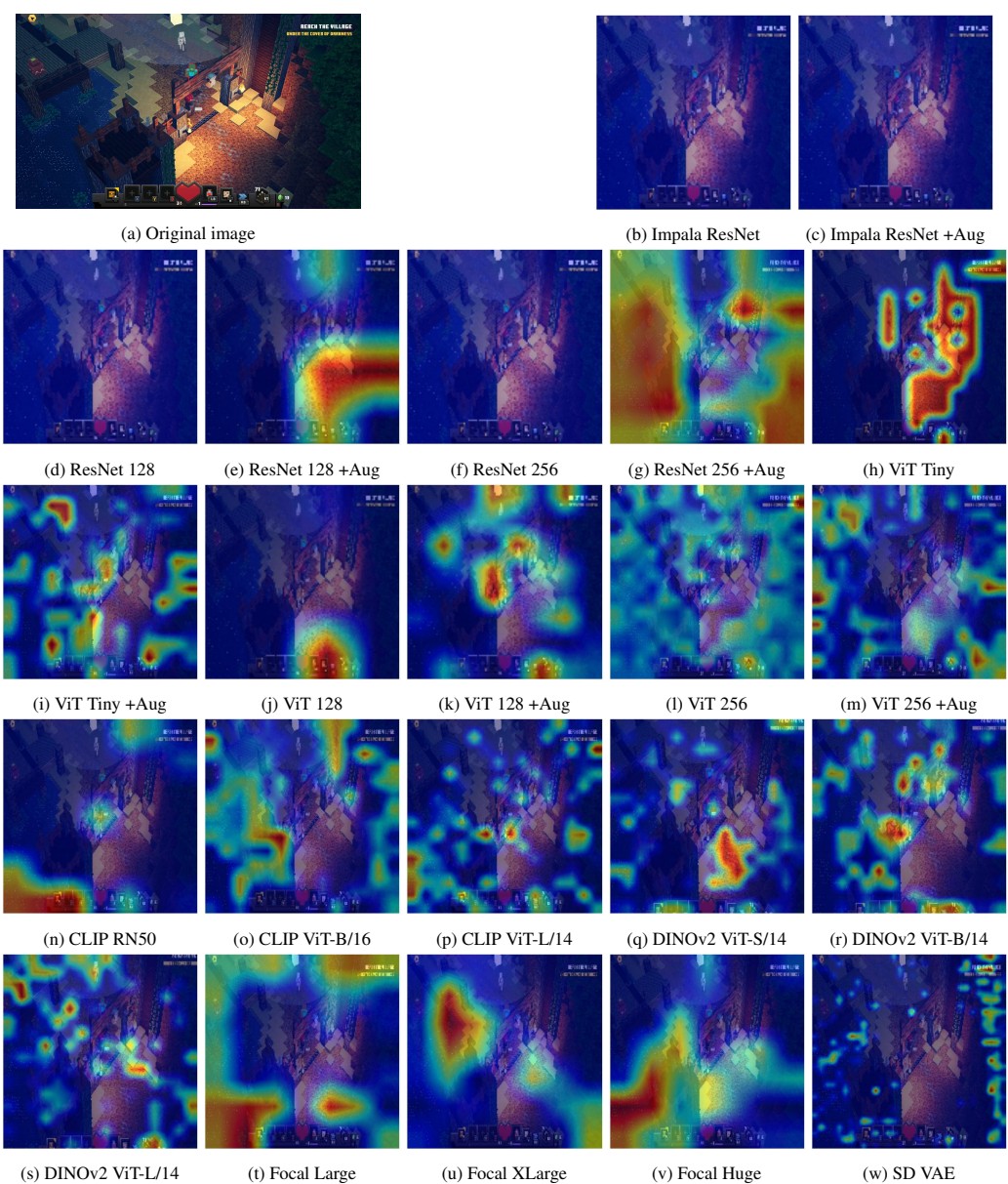

(a) Original image          (b) Impala ResNet    (c) Impala ResNet +Aug

(d) ResNet 128    (e) ResNet 128 +Aug    (f) ResNet 256    (g) ResNet 256 +Aug    (h) ViT Tiny

(i) ViT Tiny +Aug    (j) ViT 128    (k) ViT 128 +Aug    (l) ViT 256    (m) ViT 256 +Aug

(n) CLIP RN50    (o) CLIP ViT-B/16    (p) CLIP ViT-L/14    (q) DINOv2 ViT-S/14    (r) DINOv2 ViT-B/14

(s) DINOv2 ViT-L/14    (t) Focal Large    (u) Focal XLarge    (v) Focal Huge    (w) SD VAE

Figure 11: Grad-Cam visualisations for all encoders (seed 0) with policy action logits serving as the targets.

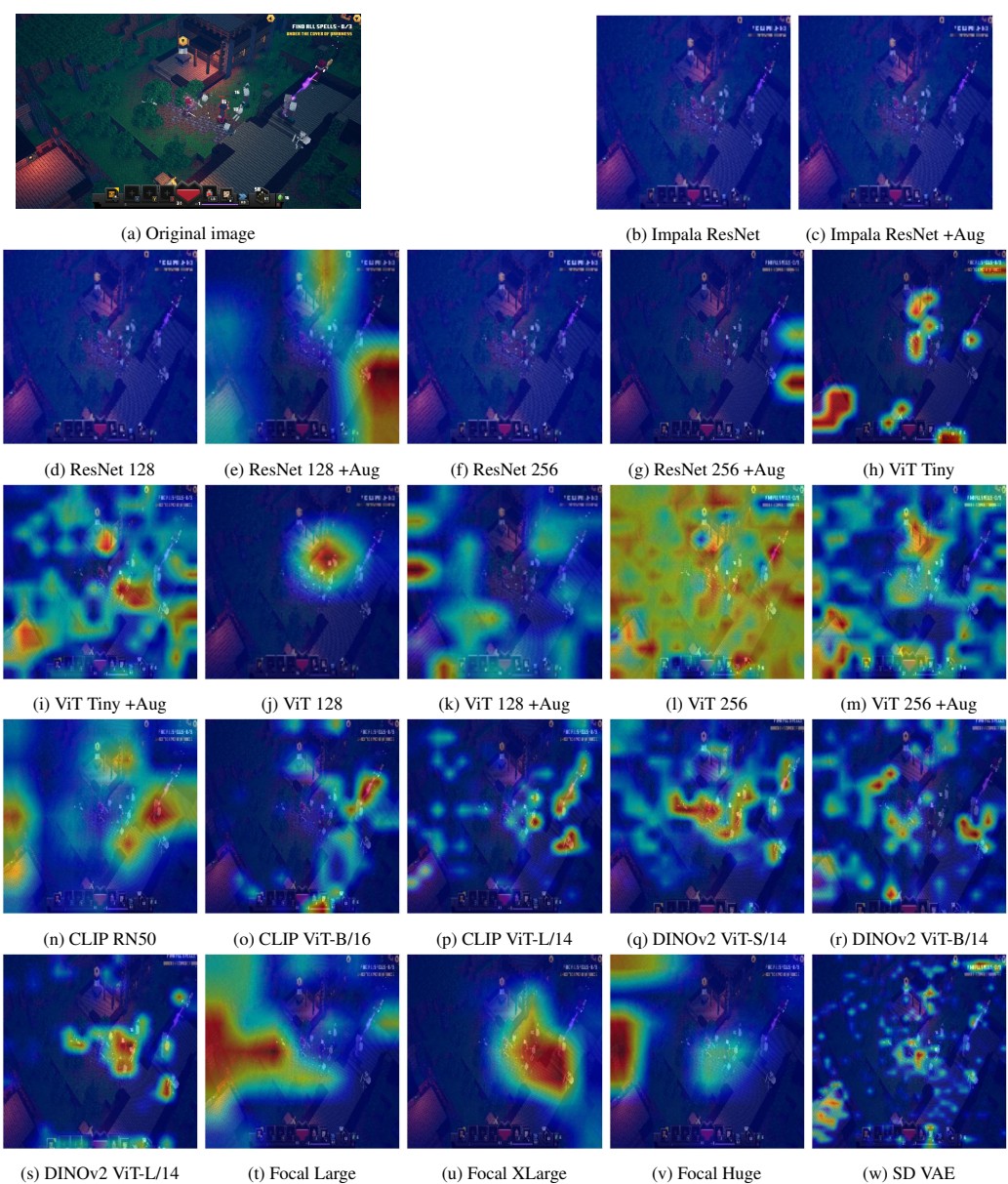

Figure 12: Grad-Cam visualisations for all encoders (seed 0) with policy action logits serving as the targets.

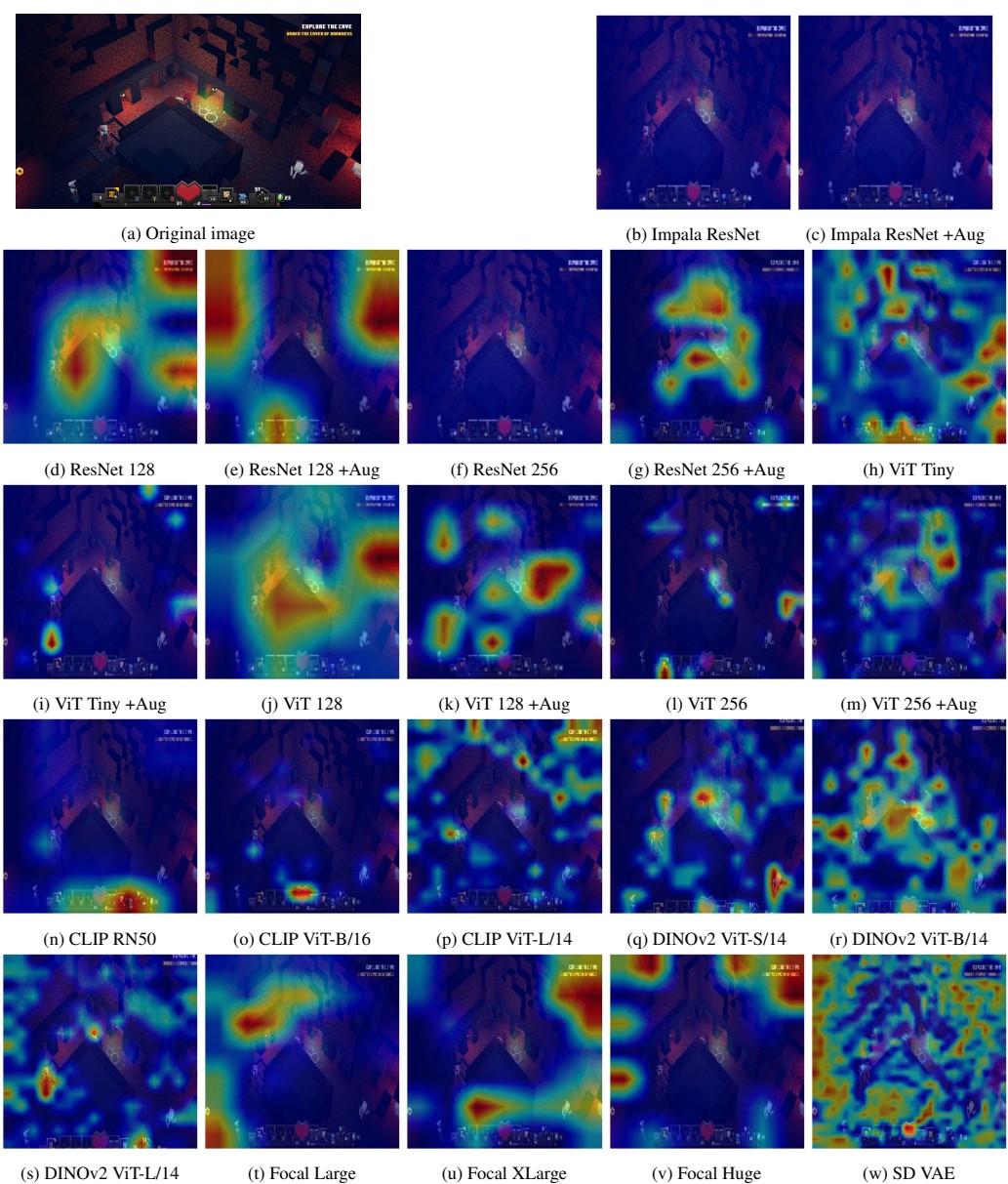

Figure 13: Grad-Cam visualisations for all encoders (seed 0) with policy action logits serving as the targets.

## F.2 MINECRAFT

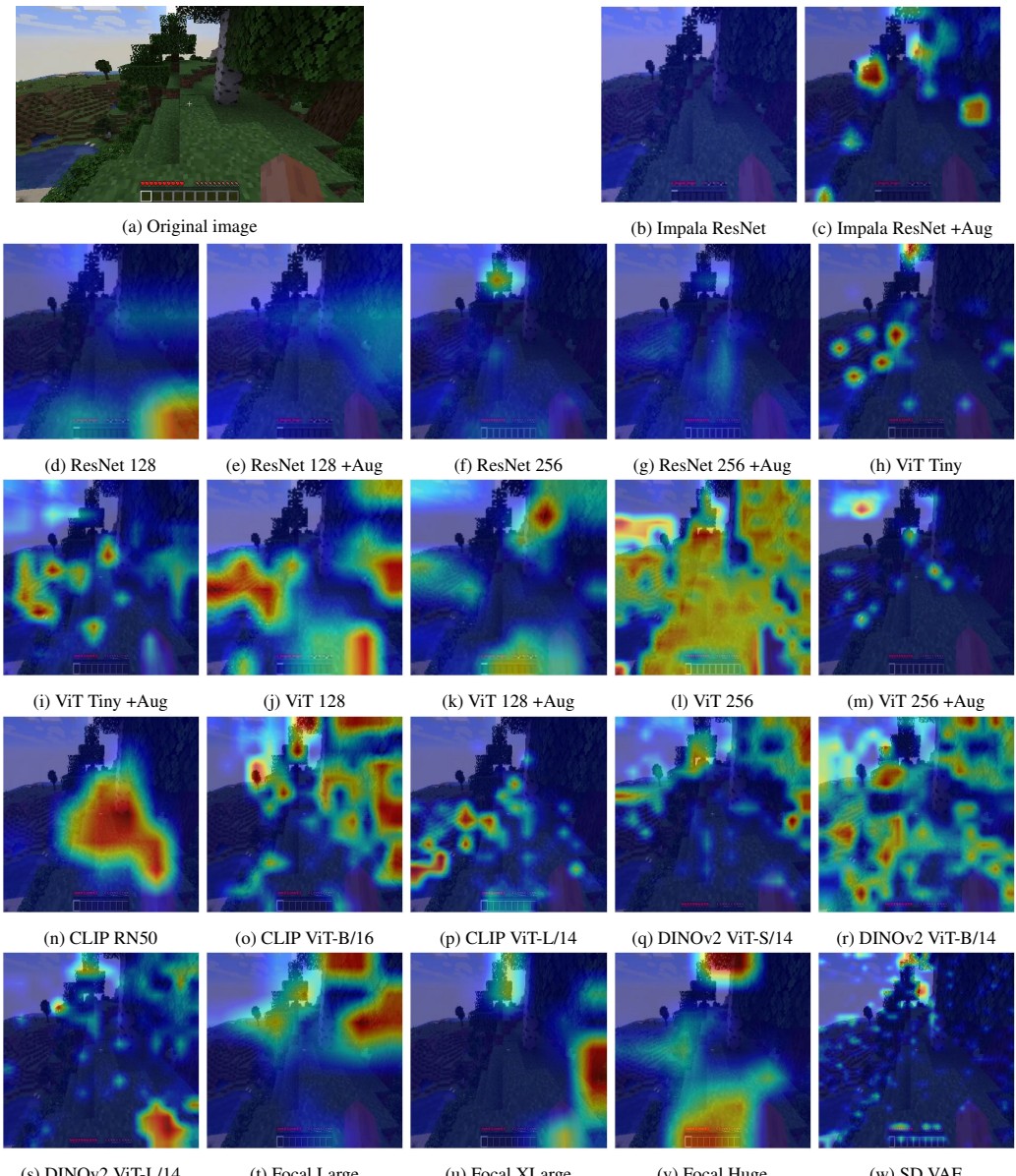

(a) Original image     (b) Impala ResNet     (c) Impala ResNet +Aug

(d) ResNet 128    (e) ResNet 128 +Aug    (f) ResNet 256    (g) ResNet 256 +Aug    (h) ViT Tiny

(i) ViT Tiny +Aug    (j) ViT 128    (k) ViT 128 +Aug    (l) ViT 256    (m) ViT 256 +Aug

(n) CLIP RN50    (o) CLIP ViT-B/16    (p) CLIP ViT-L/14    (q) DINOv2 ViT-S/14    (r) DINOv2 ViT-B/14

(s) DINOv2 ViT-L/14    (t) Focal Large    (u) Focal XLarge    (v) Focal Huge    (w) SD VAE

Figure 14: Grad-Cam visualisations for all encoders (seed 0) with policy action logits serving as the targets.

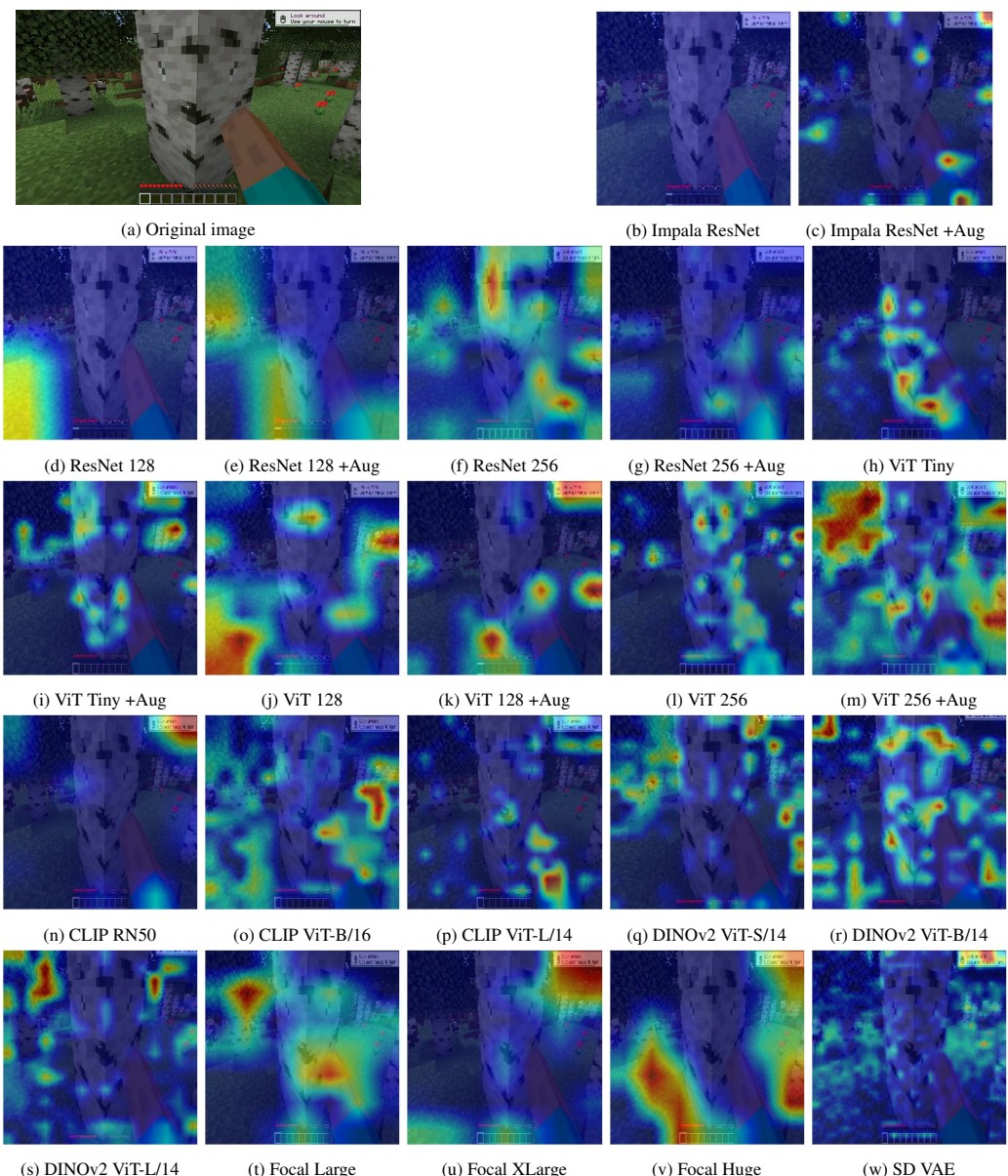

Figure 15: Grad-Cam visualisations for all encoders (seed 0) with policy action logits serving as the targets.

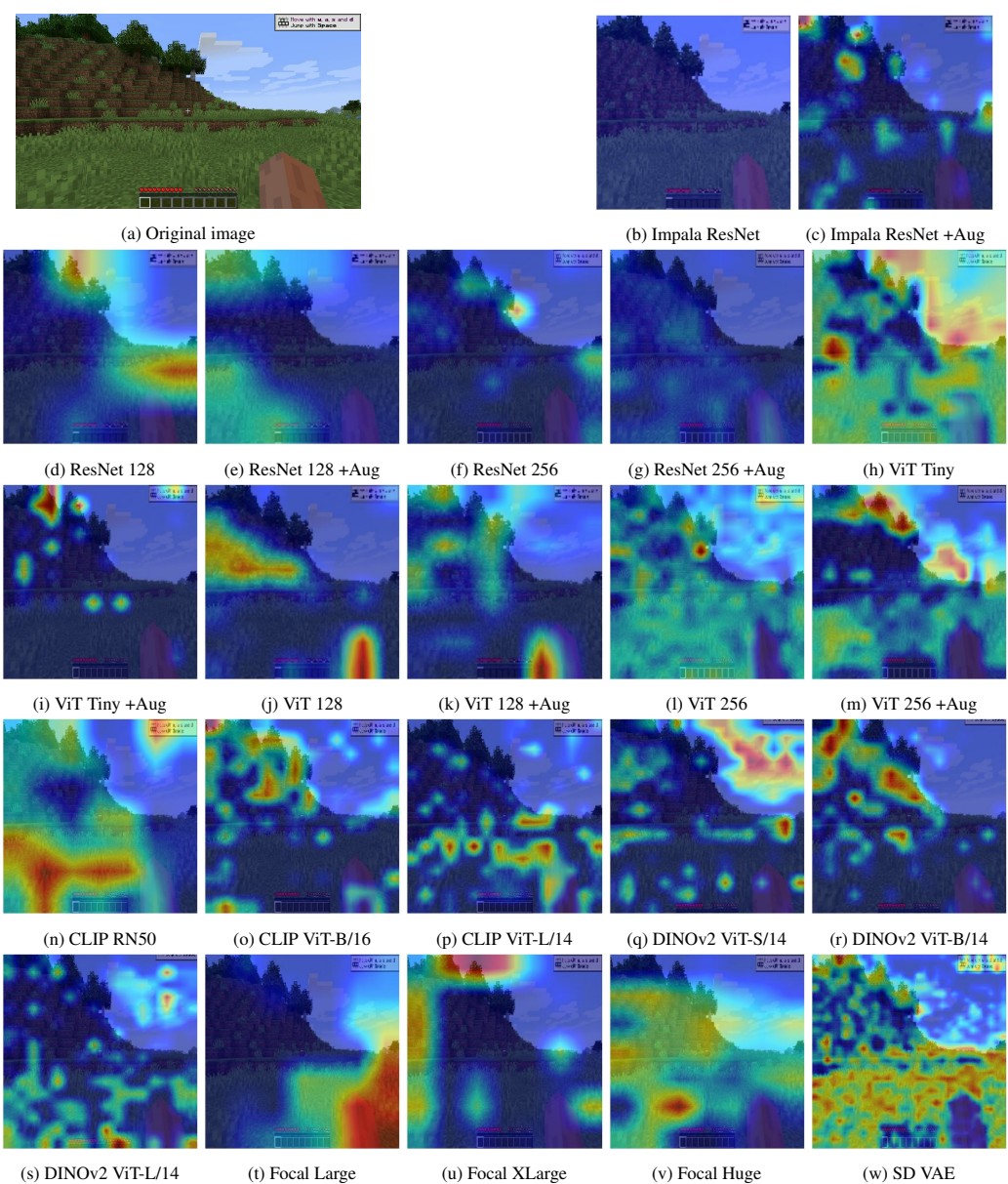

Figure 16: Grad-Cam visualisations for all encoders (seed 0) with policy action logits serving as the targets.

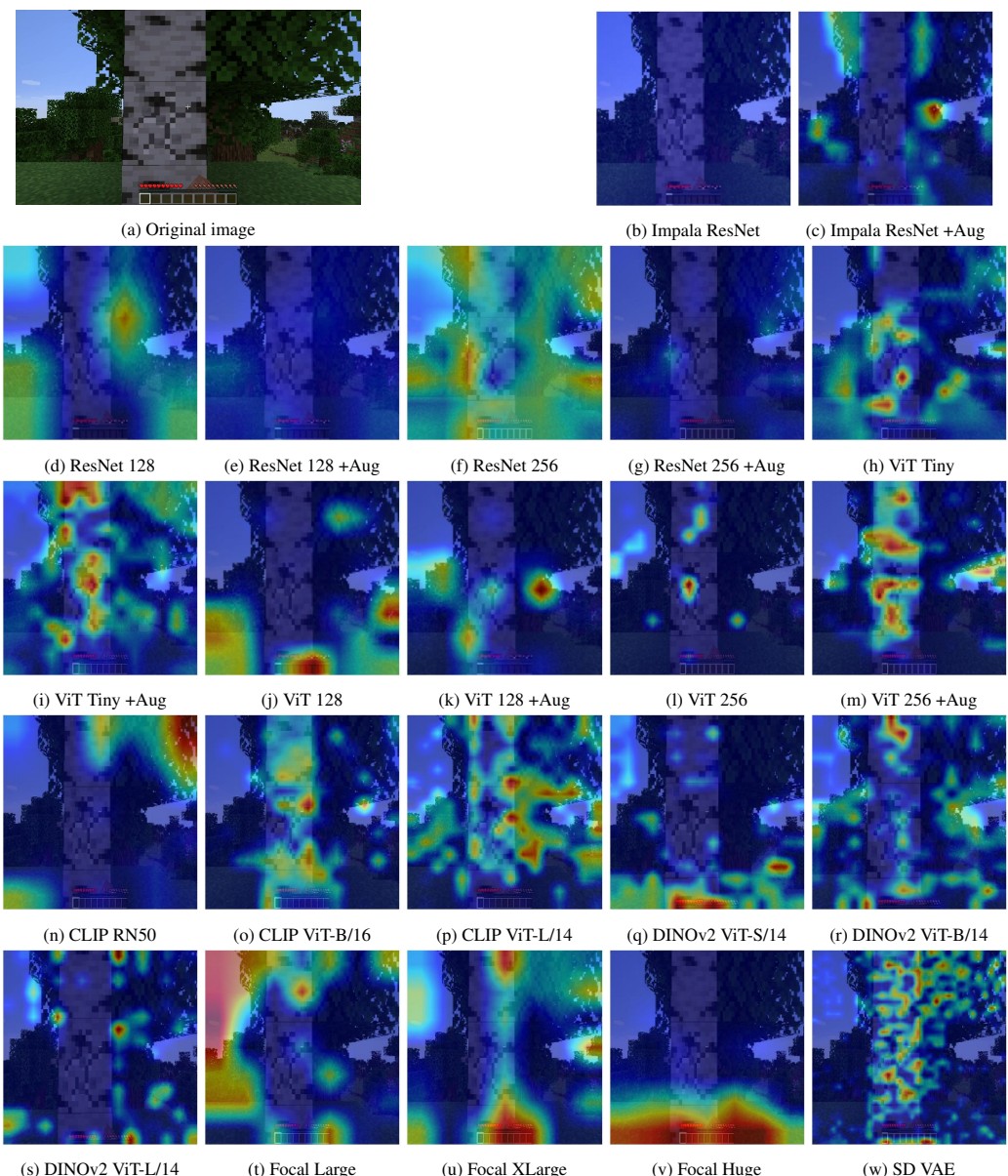

Figure 17: Grad-Cam visualisations for all encoders (seed 0) with policy action logits serving as the targets.

## G  TRAINING AND EVALUATION HARDWARE

All training runs have been completed using Azure compute using a mix of Nvidia 16GB V100s, 32GB V100s and A6000 GPUs.

**Minecraft Dungeons**   For Minecraft Dungeons, end-to-end training runs for Impala ResNet, custom ResNets (for $128 \times 128$ and $256 \times 256$ images) and custom ViT for $128 \times 128$ images without image augmentation have been done on four 16GB V100s for each run. Training runs for the same models with image augmentation have been run on one A6000 GPU (with 48GB of VRAM) for each run. Training the ViT Tiny and ViT model for $256 \times 256$ images needed more VRAMs, so these were trained on eight 16GB V100s for each run.

For training runs using pre-trained visual encoders, we computed the embeddings of all images in the Minecraft Dungeons dataset prior to training for more efficient training using A6000 GPUs. After, we were able to train each model using pre-trained visual encoders with four 16GB V100s for a single run.

To train models on half or a quarter of the training data for the third set of experiments, we used four 16GB V100s for a single run of any configuration.

Since the Minecraft Dungeons game is unable to run on Linux servers, we used Azure virtual machines running Windows 10 for the online evaluation. For evaluation of end-to-end trained models, we use a machine with two M60 GPUs, 24 CPU cores and 224GB of RAM. However, we noticed that this configuration was insufficient to evaluate models with larger pre-trained visual encoders at the desired 10Hz. Therefore, we used a configuration with one A10 GPU, 18 CPU cores and 220GB of RAM which was able to run the game and rollout the trained policy close to the desired 10Hz for all models.

**Minecraft**   The training hardware is similar to Minecraft Dungeons, with A6000s used for embedding/training with pretrained models, and 32GB V100s used to train the end-to-end models. Training pretrained models took considerably less time, with most models training within hours on a single A6000 GPU.

Minecraft evaluation was performed on remote Linux machines with A6000s, as MineRL is able to run on headless machines with virtual X buffers (`xvfb`). Each GPU had maximum of three rollouts happening concurrently, with each rollout running at 3-9 frames per second, depending on the model size.

**Counter-Strike: Global Offensive**   Training was performed on the same hardware as with Minecraft experiments. For evaluation, we ran CS:GO on a local Windows machine with an Nvidia Titan X, as per instructions in the original CS:GO paper Pearce & Zhu (2022). We ran the game at half speed (and adjusted action rate accordingly) to allow model to predict actions in time.

