# OpenReview forum: "Visual Encoders for Data-Efficient Imitation Learning in Modern Video Games"
_ICLR.cc/2024/Conference — Submitted to ICLR 2024_

### Official Review · Reviewer_bAy4 · 2023-10-27

**Soundness:** 2 fair
**Presentation:** 3 good
**Contribution:** 2 fair
**Rating:** 3
**Confidence:** 5

**Summary:**

This paper studied different video encoders for imitation learning in modern video games. The motivation is that existing pre-trained models are usually trained on real-world images, while the impact of distributional shift on video-game images remains unknown. The paper conducted a systematic research that compared different pre-trained visual encoders and from-scratch trained visual encoders in three video games. The observations suggest that pre-trained self-supervised models are worth trying in video game agent development.

**Strengths:**

- The writing is brilliant. The paper is very easy to follow.
- The study is systematic and leads to some interesting observations. The paper also gives insightful analysis for these observations, which may shed some light on the research of video-game agent development.
- The motivation is clear, and the identified problem (video-game image distribution is different from pre-training distribution) is meaningful for the community.

**Weaknesses:**

- Although the paper offered many insights and potential analysis from the emprical observations, the paper lacks enough decisive conclusions. To be specific, I find that the following claims are not convincing:
  - In the last sentence from section 5.1: "while ViTs do not guarantee improvement over ResNets, they can provide significant improvement." This conclusion is drawn from the observation that ResNets are comparable with ViTs in Minecraft Dungeons, while are outperformed by ViTs for a large margin in Minecraft. However, the observation in Table 4 (the experiments in CS:GO) shows that ResNet outperforms ViT significantly. Therefore, it still remains unknown which of these two types of networks should be chosen as visual encoder.
  - In section 5.3, "This finding suggests that, if high-quality data is available for the specific task, it might be beneficial to consider training visual encoders end-to-end for BC agents, even in situations with less available data." As the finding shows the end-to-end encoders is comparable to pre-trained encoders, why do you say that end-to-end is beneficial? Moreover, as the pre-trained backbone is fixed during imitation learning in the paper, the trainable parameters for pre-trained settings are significantly less than end-to-end setting. I am curious if the performance will be better or worse if we don't fix the pre-trained visual encoders.
  - In the last paragraph of section 5.5, it states that the pre-trained visual encoders fail to generalize when the input image size shifts. There are three related questions:
    - The resize operation seems unreasonable. As the pre-trained encoder is fixed during BC training, the feature extracted from the image is fixed, which is distorted during resizing. What about padding the image to 280x280 and then resize it to 224x224?
    - How about unfreezing the visual encoders during training? It may address the last point I raised as the visual encoder can adapt to new input size during fine-tuning.
    - The conclusion is drawn from only one experiment, how about other cases that the pre-trained model fails when input image sizes are different?

- The conclusions are drawn without controlling some critical variables. For example, the effect of network size is overlooked in the paper.
- The experiments are constrained in a limited number of video game tasks. For example, there are thousands of tasks in Minecraft as shown by [1], and this paper only tested on the "Treechop" task. Also, the paper only studied the task-specific imitation learning, while large-scale pre-training adopted by VPT [2] or multi-task imitation learning [3] are not examined.
- (minor) A key motivation of this paper is that, the images are often related to real-world scenes, which differs from video games. But there seems not enough suppotive evidence in the paper, and the readers usually don't know whether video game images are used during pre-training. Maybe a summary table that presents the pre-training sources of different models will be clear.



[1] Minedojo: Building open-ended embodied agents with internet-scale knowledge. In NeurIPS, 2022.

[2] Video pretraining (vpt): Learning to act by watching unlabeled online videos. In NeurIPS, 2022.

[3] Open-World Multi-Task Control Through Goal-Aware Representation Learning and Adaptive Horizon Prediction. In CVPR 2022.

**Questions:**

- How do you select the hyper-parameters for different experiments?
- What are the original image sizes for Minecraft Dungeons and Minecraft?
- Why does the paper use FocalNet as classification supervised pre-trained encoders? ImageNet pre-trained models such as ViT / DeiT are also popular, which share the same architectures as in the other categories (language contrastive / self-supervised pre-trained) and thus are easy to be compared.

---

> ### Author Response · Authors · 2023-11-16
> **Rebuttal to Reviewer bAy4 (1/2)**
>
> We thank the reviewer for their comments and suggestions. Below, we will address the main questions and concerns raised by the reviewer.
>
> > How do you select the hyper-parameters for different experiments?
>
> We refer the reviewer to Section “Hyperparameter Selection” in our common response where we clarify how hyperparameters were selected.
>
> > What are the original image sizes for Minecraft Dungeons and Minecraft?
>
> Our collected data in Minecraft Dungeons and the filtered dataset in Minecraft contain images of resolutions 1280x720 and 640x360, respectively. We added this information to the environment descriptions in the paper.
>
> > Why does the paper use FocalNet as classification supervised pre-trained encoders? ImageNet pre-trained models such as ViT / DeiT are also popular, which share the same architectures as in the other categories (language contrastive / self-supervised pre-trained) and thus are easy to be compared.
>
> We agree that there are popular and well-performing ViT (and ViT variations) models trained on ImageNet classification. While these models are architecturally more similar to other pre-trained and end-to-end trained models, we decided to instead include FocalNet since these models have recently been proposed and have been shown to outperform ViT-based models in the ImageNet benchmark [1].
>
> > Although the paper offered many insights and potential analysis from the emprical observations, the paper lacks enough decisive conclusions.
>
> We thank the reviewer for their detailed thoughts. We revised the description of the discussed conclusions and point the reviewer to the colour-coded parts in the revised paper.
>
> > The conclusions are drawn without controlling some critical variables. For example, the effect of network size is overlooked in the paper.
>
> End-to-end visual encoders mostly have a comparable number of parameters and therefore do not allow clear conclusions about network size, but we evaluate three models of varying sizes for CLIP, DINOv2 and FocalNet. For the model size of these pre-trained encoders, we do not observe any clear relationship with online performance. While larger DINOv2 models perform best in Minecraft, the same trend does not hold for CLIP and FocalNet where encoders with fewer parameters perform better. Results for Minecraft Dungeons do not allow any clear conclusions either. The same inconclusive trend continues for pre-trained visual encoders when reducing the amount of training data. Therefore, at least for pre-trained visual encoders, we can confidently say that the number of parameters does not seem to be a clear indicator for online evaluation performance. We included a brief description of these conclusions to our revised paper (see colour-coded sentences).
>
> > How about unfreezing the visual encoders during training? It may address the last point I raised as the visual encoder can adapt to new input size during fine-tuning.
>
> We refer the reviewer to Section “Fine-tuning of Pre-Trained Visual Encoders” in our common response. In short, fine-tuning pre-trained visual encoders would come at a considerable computational cost that is not feasible at the breadth of our empirical study.
>
> > In the last paragraph of section 5.5, it states that the pre-trained visual encoders fail to generalize when the input image size shifts. …
>
> We agree with the reviewer that our experiments in CS:GO allow no clear conclusions as to why pre-trained visual encoders fail. To investigate our hypothesis that the image processing might be detrimental to DINOv2, we compare the performance of four pre-trained visual encoders (one for each of the four families CLIP, DINOv2, FocalNet, Stable Diffusion) with three different types of image processing in Minecraft. For details, we refer the reviewer to Appendix E which describes the conducted experiment and findings. In summary, we do not find any evidence that image processing, as applied in CS:GO, is detrimental to the performance of pre-trained visual encoders in Minecraft. This raises the open question of why pre-trained DINOv2 visual encoders are not effective for decision making in CS:GO, in contrast to Minecraft Dungeons and Minecraft. We hope this paper raises awareness of this unique challenge to the ICLR community and will open source all required code to reproduce and build on this result.
>
> > For example, there are thousands of tasks in Minecraft as shown by [1], and this paper only tested on the "Treechop" task.
>
> We acknowledge the simplicity of the Treechop task in Minecraft but would like to highlight that the goal of our study is to cover a range of video games. We refer the reviewer to Section “Evaluation Tasks” in our common response where we discuss the selection of tasks more.

---

> > ### Author Response · Authors · 2023-11-16
> > **Rebuttal to Reviewer bAy4 (2/2)**
> >
> > > Also, the paper only studied the task-specific imitation learning, while large-scale pre-training adopted by VPT [2] or multi-task imitation learning [3] are not examined.
> >
> > Regarding the suggestion to leverage large-scale pre-training as adopted by VPT, we do not believe that such pre-training would be feasible for evaluation study in which we cover a total of 22 different visual encoder configurations across three complex evaluation environments. Regarding the multi-task imitation learning suggestion, we thank the reviewer for pointing us to this concurrent work which we now include in Section 2 on related work.
> >
> > For a more detailed discussion on the relationship between both mentioned work and our work, we refer the reviewer to Section “Discussion of Related Work” in our common response.
> >
> > > (minor) A key motivation of this paper is that, the images are often related to real-world scenes, which differs from video games. But there seems not enough suppotive evidence in the paper, and the readers usually don't know whether video game images are used during pre-training. Maybe a summary table that presents the pre-training sources of different models will be clear.
> >
> > We agree with the reviewer that understanding the pre-training sources of different models is highly relevant for our study. Unfortunately, this information is not publicly available for OpenAI CLIP and Stable Diffusion pre-trained visual encoders to the best of our knowledge. The WebImageText dataset, collected to train OpenAI CLIP, is stated to contain 400 million (image, text) pairs collected form a variety of publicly available sources on the Internet with the collection process prioritising text which commonly occurs in the English version of Wikipedia (see Section 2.2 in [2]). However, it is not publicly available what exact information has been used to train OpenAI CLIP.
> >
> > For FocalNet, we leverage visual encoders pre-trained on the classification dataset of ImageNet 22K which contains real-world images only (no video game images) [1].
> >
> > For DINOv2, the training dataset is stated to contain 142 million images collected from a range of publicly available image datasets for classification, fine-grained classification, segmentation, depth estimation, and retrieval (see Table 15 in [3]). We highlight that all these datasets contain real-world images. This suggests that DINOv2 models have never been trained on images representative of video games, suggesting that despite a significant distribution shift from its training data to our evaluation domains, DINOv2 visual encoders effectively generalise to modern video games.
> >
> > ## Citations
> > [1] Yang, Jianwei, Chunyuan Li, Xiyang Dai, and Jianfeng Gao. "Focal modulation networks." Advances in Neural Information Processing Systems 35 (2022): 4203-4217.
> >
> > [2] Radford, Alec, Jong Wook Kim, Chris Hallacy, Aditya Ramesh, Gabriel Goh, Sandhini Agarwal, Girish Sastry et al. "Learning transferable visual models from natural language supervision." In International conference on machine learning, pp. 8748-8763. PMLR, 2021.
> >
> > [3] Oquab, Maxime, Timothée Darcet, Théo Moutakanni, Huy Vo, Marc Szafraniec, Vasil Khalidov, Pierre Fernandez et al. "Dinov2: Learning robust visual features without supervision." arXiv preprint arXiv:2304.07193 (2023).

---

> > > ### Comment · Reviewer_bAy4 · 2023-11-20
> > >
> > > Dear Authors,
> > >
> > > Thank you for your comprehensive responses. However, I find the scope of the conclusions in this paper to be overly narrow, and the necessity to exclude numerous outlier scenarios is concerning. The conclusions drawn from the experiments lack universal applicability, which casts doubt on the broader relevance of the paper's contributions to the field.
> > >
> > > Regarding the conclusion section:
> > >
> > > The claim that "training visual encoders end-to-end on relatively small images can yield strong performance when using high-quality, representative data for the evaluation task, even in low-data regimes of few hours or minutes" seems overly optimistic. This assertion appears valid only for simpler tasks that do not demand high-resolution perception. The study's focus on the TreeChop task in Minecraft fails to recognize the limitations of low-resolution visuals in more complex tasks, such as crafting items, where finer details are crucial for accurate item identification.
> > >
> > > The statement "DINOv2, trained with self-supervised objectives on diverse data, consistently outperformed other pre-trained visual encoders, indicating its generality and suitability for video games" is problematic. For instance, DINOv2's performance in CS:GO was notably poor, a fact acknowledged by the authors. This inconsistency raises questions about the robustness of DINOv2 across different gaming environments.
> > >
> > > The paper's limited focus on just three video games undermines the reliability of its conclusions. Specifically, the inconsistency observed in CS:GO challenges the generalizability of the findings to other video games. This limitation should be addressed to enhance the credibility and applicability of the research.
> > >
> > > In summary, the paper's contributions seem constrained by its narrow focus and the exclusion of critical scenarios, diminishing its impact and relevance to the broader community. More comprehensive testing and analysis, particularly in diverse and challenging environments, would be necessary to strengthen the conclusions and enhance the paper's contribution to the field.

---

### Official Review · Reviewer_sq4s · 2023-10-29

**Soundness:** 2 fair
**Presentation:** 3 good
**Contribution:** 1 poor
**Rating:** 1
**Confidence:** 5

**Summary:**

This work compares using pretrained image encoders with learned end-to-end encoders trained with behavioral cloning. They consider a few variants of both ResNets and ViT end-to-end encoders with and without image augmentation. They compare these to encoders from language contrastive pretraining (CLIP), self-supervised pretraining (DINOv2), supervised pretraining (FocalNet), and reconstruction based pretraining (VAE). They compare this methods in 3 modern video game settings: Minecraft Dungeons, Minecraft, and Counter-Strike GO.

They find that image augmentation improves performance for end-to-end BC encoders in some cases, but in other cases it is better to train end-to-end. They first compare which end-to-end encoder is best and find ViT’s to be the most performant. They find that amongst the considered pretrained encoders, DINOv2 performed best. They further compare these methods in more data limited regimes; surprisingly, results are mixed even in the data-limited regime where one would expect pretrained encoders to shine.

**Strengths:**

The authors provide a valuable datapoint to the community for which existing pretrained encoders they may want to initialize their experiments from (seemingly DINO).

**Weaknesses:**

Small scope and unsurprising results. This paper is more of a baselines paper comparing existing methods. For a baselines paper, I would expect far more extensive experiments across domains and methods.

The domains considered here, while they are “modern video games”, are quite limited. E.g. for Minecraft they only consider the treechop task, which is the most basic thing one can do in Minecraft

**Questions:**

How do these methods compare in more domains? I would also expect experiments in simpler domains like e.g. atari, coinrun, maybe robotics environments.

How do the methods considered here compare to other common methods?

- auxiliary objectives for representation learning are quite common in reinforcement learning. Given this paper is studying the efficacy of different image encoders, it would seem natural to me to also include auxiliary self-supervised objectives into the end-to-end experiments
- the authors hold pretrained encoders fixed. It seems natural to also comp

How well tuned were the experiments for different encoders? It seems hyperparameters were held fixed across all architectures.

---

> ### Author Response · Authors · 2023-11-16
> **Rebuttal to Reviewer sq4s (1/2)**
>
> We thank the reviewer for their comments and suggestions. Below, we will address the main comments raised by the reviewer.
>
> > This paper is more of a baselines paper comparing existing methods. For a baselines paper, I would expect far more extensive experiments across domains and methods.
>
> We would like to highlight that our experiments evaluate a total of 22 different visual encoder configurations across two tasks, each model being evaluated across three seeds in each task, with a partial set of visual encoders being considered in a third task. We believe our evaluation already covers a wide range of commonly used visual encoders. Furthermore, we outline our reasoning for the selection of existing tasks and their complexity in Section “Evaluation Tasks” of our common response.
>
> > The domains considered here, while they are “modern video games”, are quite limited. E.g. for Minecraft they only consider the treechop task, which is the most basic thing one can do in Minecraft
>
> and
>
> > How do these methods compare in more domains? I would also expect experiments in simpler domains like e.g. atari, coinrun, maybe robotics environments.
>
> We acknowledge the simplicity of the Treechop task in Minecraft but would like to highlight that the goal of our study is to cover a broad range of modern video games. We refer the reviewer to Section “Evaluation Tasks” in our common response where we discuss the selection of tasks more.
>
> Regarding the suggestion to run simpler domains like Atari or Coinrun, we would like to highlight that, first, the focus of our work is on modern video games that go beyond Atari (as stated in the abstract). Second, Atari and Coinrun both offer a programmatic interface and are therefore common environments for reinforcement learning. In contrast, our work focuses on imitation learning in video games where no such interface is available. Third, imitation learning in Atari is already a well-studied problem with a plethora of prior work [1-6]. For Coinrun, few work consider the application of imitation learning due to the lack of datasets of (human) demonstrations. Existing work therefore trains policies from demonstrations collected by reinforcement learning agents [7] which does not align with our focus on imitating human gameplay.
>
> Lastly, our work already acknowledges and cites an existing empirical study on the application of imitation learning with varying visual encoders in robotics environments [8]. Given this study, we focus on the settings of video games which are meaningfully different from robotics since robotics tasks often represent the current state using images akin to the real world. In contrast, video games often have highly stylised visual representations (e.g. Minecraft and Minecraft Dungeons) which are notably different from the real-world images many pre-trained visual encoders are trained on. This represents a significant shift in distribution of encountered images which raised the research question whether visual encoders pre-trained on real-world images can be effective in video games. Our study clearly answers this question affirmatively and provides clear evidence as to which pre-trained visual encoders should be considered for application in complex video games.
>
>
> > auxiliary objectives for representation learning are quite common in reinforcement learning. Given this paper is studying the efficacy of different image encoders, it would seem natural to me to also include auxiliary self-supervised objectives into the end-to-end experiments
>
> As the reviewer rightfully states, these auxiliary objectives are commonly applied in reinforcement learning. However, we train agents purely with imitation learning in domains where no reinforcement learning signal is available. In our evaluation domains, there exists no or only a limited programmatic interface which does not provide all necessary information used in many of these reinforcement learning auxiliary objectives so these approaches are not applicable to our setting or have not been established yet to serve as additional training objectives in our setting.
>
> > the authors hold pretrained encoders fixed. It seems natural to also comp
>
> We infer that the reviewer meant to suggest to fine-tune pre-trained visual encoders. In this case, we refer the reviewer to Section “Fine-tuning of Pre-Trained Visual Encoders” of our common response in which we illustrate why we have not considered fine-tuning pre-trained visual encoders in our study. We kindly ask the reviewer to clarify if they meant to make a different suggestion.

---

> > ### Author Response · Authors · 2023-11-16
> > **Rebuttal to Reviewer sq4s (2/2)**
> >
> > > How well tuned were the experiments for different encoders? It seems hyperparameters were held fixed across all architectures.
> >
> > We refer the reviewer to Section “Hyperparameter Selection” in our common response where we clarify how hyperparameters were selected.
> >
> > ## Citations
> > [1] Kurin, Vitaly, Sebastian Nowozin, Katja Hofmann, Lucas Beyer, and Bastian Leibe. "The atari grand challenge dataset." arXiv preprint arXiv:1705.10998 (2017).
> >
> > [2] Zhang, Ruohan, Calen Walshe, Zhuode Liu, Lin Guan, Karl Muller, Jake Whritner, Luxin Zhang, Mary Hayhoe, and Dana Ballard. "Atari-head: Atari human eye-tracking and demonstration dataset." In Proceedings of the AAAI conference on artificial intelligence, vol. 34, no. 04, pp. 6811-6820. 2020.
> >
> > [3] Yu, Xingrui, Yueming Lyu, and Ivor Tsang. "Intrinsic reward driven imitation learning via generative model." In International conference on machine learning, pp. 10925-10935. PMLR, 2020.
> >
> > [4] Reddy, Siddharth, Anca D. Dragan, and Sergey Levine. "Sqil: Imitation learning via reinforcement learning with sparse rewards." International Conference on Learning Representations (2020).
> >
> > [5] Chen, Brian, Siddhant Tandon, David Gorsich, Alex Gorodetsky, and Shravan Veerapaneni. "Behavioral cloning in atari games using a combined variational autoencoder and predictor model." In 2021 IEEE Congress on Evolutionary Computation (CEC), pp. 2077-2084. IEEE, 2021.
> >
> > [6] Kanervisto, Anssi, Joonas Pussinen, and Ville Hautamäki. "Benchmarking end-to-end behavioural cloning on video games." In 2020 IEEE conference on games (CoG), pp. 558-565. IEEE, 2020.
> >
> > [7] Chen, Xin, Sam Toyer, Cody Wild, Scott Emmons, Ian Fischer, Kuang-Huei Lee, Neel Alex et al. "An empirical investigation of representation learning for imitation." NeurIPS, Track on Datasets and Benchmarks (2021).
> >
> > [8] Nair, Suraj, Aravind Rajeswaran, Vikash Kumar, Chelsea Finn, and Abhinav Gupta. "R3m: A universal visual representation for robot manipulation." Conference on Robot Learning (2022).

---

### Official Review · Reviewer_JgYZ · 2023-10-30

**Soundness:** 2 fair
**Presentation:** 3 good
**Contribution:** 2 fair
**Rating:** 3
**Confidence:** 5

**Summary:**

This paper studies an important problem: whether a pre-trained vision encoder can boost the performance of sequential decision-making models. The authors comprehensively study four primary encoder categories: self-supervised trained, supervised trained, contrastive-learning trained, and reconstruction trained, and draw several interesting conclusions. This will be meaningful for choosing backbones to design policy models in complicated environments.

**Strengths:**

* The paper is well-written and easy to follow.
* This paper studies an important problem: the difference of vision encoders in building policy models for decision-making.
* The selected environments are three modern video games, which are popular and challenging. To some degree, I believe the conclusions drawn from these environments can be generalized to real-world scenarios.

**Weaknesses:**

* **Missing some details.** It is not clear what kinds of image augmentation tricks are used. Why the image augmentation method is specific to the game? Why a pre-trained model (DINOv2) is better than the others? It lacks deep discussions.

* **Provides rollout videos for better understanding.** Rollout videos are very helpful for readers to understand the challenges of the environments and the effectiveness of the model. It is strongly recommended to include some videos in the supplementary materials.

* **Insufficient evaluation tasks in the Minecraft domain.** In Minecraft, the "Treechop" task is the most basic and simple task. Although it is an important benchmark, however, conducting experiments solely on this task is not enough. It is better to include 2-3 extensive tasks, such as "Hunt animals", "Craft crafting_tables", and "Mine ores", to enhance the soundness.

* **Concerns about the training data distribution of baselines.**

* **Missing some baselines and references.** [1] proposed an important foundation model for decision-making in Minecraft, which was trained on large-scale YouTube gameplays with behavior cloning. It yields a good vision encoder that is specified in the Minecraft domain. Although it is cited in the paper, it does not participate in the comparison. I suggest the authors to compare VPT in the experiment. [2] is a large-scale pre-trained segmentation model, which has demonstrated strong cross-domain recognition capability. It should be included as a baseline. [3, 4, 5] are also imitation learning methods in the Minecraft domain, which are strongly related to this topic. I suggest the author reference these works and have necessary discussions.

[1] "Video PreTraining (VPT): Learning to Act by Watching Unlabeled Online Videos", https://arxiv.org/abs/2206.11795

[2] "Segment anything", https://arxiv.org/abs/2304.02643

[3] "STEVE-1: A Generative Model for Text-to-Behavior in Minecraft", https://arxiv.org/abs/2310.08235

[4] "Open-World Multi-Task Control Through Goal-Aware Representation Learning and Adaptive Horizon Prediction", https://arxiv.org/abs/2301.10034

[5] "GROOT: Learning to Follow Instructions by Watching Gameplay Videos", https://arxiv.org/abs/2310.08235

**Questions:**

My questions are listed in the weakness part.

I will consider improving the rating if the author adequately addresses my concerns.

---

> ### Author Response · Authors · 2023-11-16
> **Rebuttal to Reviewer JgYZ (1/2)**
>
> We thank the reviewer for their comments and suggestions. Below, we will address the main comments raised by the reviewer.
>
> > It is not clear what kinds of image augmentation tricks are used
>
> As stated in Section 3.2, we apply the same image augmentations also applied by VPT [1]. To further clarify, we added a description of the applied image augmentations in Appendix A.1 in the revised paper.
>
> > Why the image augmentation method is specific to the game?
>
> We apply the same image augmentation in each game. We find image augmentations to be effective in Minecraft Dungeons and harmful to training in Minecraft which might be surprising but is consistent with prior findings where image augmentations can significantly improve the robustness of decision making agents [2, 3] but also harm performance if it perturbs images too much [4].
>
> > Why a pre-trained model (DINOv2) is better than the others? It lacks deep discussions.
>
> We acknowledge that from our experiments, it is difficult to identify the reason why some pre-trained models work better than others. We hypothesise that the self-supervised training objective of DINOv2 leads to more general embeddings than reconstruction (tries to recreate all pixels rather than embed high-level information), classification (tries to only embed information related to predicted ImageNet classes), and language-contrastive learning (embed information correlated with language). This might be particularly important in domains, such as video games, which both visually and conceptually differ notably from the real-world images these models are typically trained on. Language descriptions, classification labels, and reconstruction might be too focused on embedding concepts important in the real world, self-supervised objectives might be more agnostic and, thus, effective in domains which differ from the real world such as video games.
>
> > Provides rollout videos for better understanding. Rollout videos are very helpful for readers to understand the challenges of the environments and the effectiveness of the model. It is strongly recommended to include some videos in the supplementary materials.
>
> We thank the reviewer for the suggestion to release rollout videos of agents. These videos can now be accessed at the following anonymous repository: https://anonymous.4open.science/r/imitation_learning_in_modern_video_games_rollouts-4C48/README.md
>
> > Insufficient evaluation tasks in the Minecraft domain. In Minecraft, the "Treechop" task is the most basic and simple task. Although it is an important benchmark, however, conducting experiments solely on this task is not enough. It is better to include 2-3 extensive tasks, such as "Hunt animals", "Craft crafting_tables", and "Mine ores", to enhance the soundness.
>
> We acknowledge the simplicity of the Treechop task in Minecraft but would like to highlight that 1. our goal is to cover a range of visually and conceptually diverse modern video games (breadth) rather than focusing on few challenging tasks in a single game (depth), and that 2. Minecraft Dungeons in particular represents a highly challenging decision making task. For a more detailed discussion of this topic, we refer the reviewer to the section “Evaluation Tasks” in our common response.
>
> > [1] (VPT) proposed an important foundation model for decision-making in Minecraft, which was trained on large-scale YouTube gameplays with behavior cloning. It yields a good vision encoder that is specified in the Minecraft domain. Although it is cited in the paper, it does not participate in the comparison. I suggest the authors to compare VPT in the experiment. [2] (SAM)  is a large-scale pre-trained segmentation model, which has demonstrated strong cross-domain recognition capability. It should be included as a baseline.
>
> We refer the reader to the sections “Discussion of Related Work” and “Segment Anything Pre-Trained Visual Encoder” in our common response where we contrast VPT to our evaluation setup in more detail and discuss the applicability of SAM to our setting. In short, we do not believe that comparisons to VPT would be fair or informative. For Segment Anything, we agree these models appear promising but unfortunately we found its inference cost to be too high for our tasks in which agents have to take actions in real-time.

---

> > ### Author Response · Authors · 2023-11-16
> > **Rebuttal to Reviewer JgYZ (2/2)**
> >
> > > [3, 4, 5] are also imitation learning methods in the Minecraft domain, which are strongly related to this topic. I suggest the author reference these works and have necessary discussions.
> >
> > We thank the reviewer for pointing out these recent, concurrent related works and have added these works in Section 2 on related work. We further refer the reviewer to the section “Discussion of Related Work” in our common response where we discuss the work on open-world multi-task control in more detail. Lastly, we would like to point out that the most recent version of [4] and the referenced paper [5] were made publicly available on arXiv after the ICLR submission deadline.
> >
> > > Concerns about the training data distribution of baselines.
> >
> > Would the reviewer be able to elaborate what concerns they have about the training distribution?
> >
> > ## Citations
> > [1] Baker, Bowen, Ilge Akkaya, Peter Zhokov, Joost Huizinga, Jie Tang, Adrien Ecoffet, Brandon Houghton, Raul Sampedro, and Jeff Clune. "Video pretraining (vpt): Learning to act by watching unlabeled online videos." Advances in Neural Information Processing Systems 35 (2022): 24639-24654.
> >
> > [2] Yarats, Denis, Ilya Kostrikov, and Rob Fergus. "Image augmentation is all you need: Regularizing deep reinforcement learning from pixels." In International conference on learning representations. 2020.
> >
> > [3] Andrychowicz, OpenAI: Marcin, Bowen Baker, Maciek Chociej, Rafal Jozefowicz, Bob McGrew, Jakub Pachocki, Arthur Petron et al. "Learning dexterous in-hand manipulation." The International Journal of Robotics Research 39, no. 1 (2020): 3-20.
> >
> > [4] Kanervisto, Anssi, Janne Karttunen, and Ville Hautamäki. "Playing minecraft with behavioural cloning." In NeurIPS 2019 Competition and Demonstration Track, pp. 56-66. PMLR, 2020.

---

> > ### Comment · Reviewer_JgYZ · 2023-11-20
> > **Response to Authors**
> >
> > + In Appendix A.1, are you sure that VPT used image augmentation tricks? Can you specify which page contains the descriptions of augmentation tricks in VPT?
> >
> > + Thank authors for providing rollout videos. I checked the rollout videos and found the performance really bad (did not chop any tree among the five videos). Such experiments will not convince me.
> >
> > + Although the task of "chop tree" is not simple, it is not enough to just do this task for the Minecraft environment. It would be more persuasive to at least conduct experiments related to hunting animals as well.
> >
> > + I did not ask the authors to directly compare their method with the original VPT in terms of success rate. However, researchers in the Minecraft community may be interested in how the video encoder of pre-trained VPT could be helpful for their tasks. As a research paper on which video encoder is better, it is meaningless to exclude the visual encoder of the most powerful model VPT.
> >
> > + Since the author mentions that the SAM speed is particularly slow, this experiment can be skipped.

---

> > > ### Author Response · Authors · 2023-11-20
> > > **VPT Augmentations**
> > >
> > > Happy to clarify with respect to VPT augmentations.
> > >
> > > The augmentations applied are described in Appendix D.2 on page 24 here: https://arxiv.org/pdf/2206.11795.pdf
> > >
> > > The section describes training details for the IDM, but as outlined in E.4, the training of the agent model follows the same specifications (besides a small detail unrelated to augmentations):
> > > > The foundation model training is similar to the IDM training, with the exception of labels being IDM-generated pseudo labels.

---

### Official Review · Reviewer_yJPq · 2023-10-31

**Soundness:** 2 fair
**Presentation:** 2 fair
**Contribution:** 2 fair
**Rating:** 3
**Confidence:** 4

**Summary:**

The paper focuses on the challenge of training agents in modern video games, going beyond simpler games like those on Atari. The central research question is: How can images be encoded for data-efficient imitation learning in modern video games? To address this, the authors compare both end-to-end trained visual encoders and pre-trained visual encoders across three modern video games: Minecraft, Minecraft Dungeons, and Counter-Strike: Global Offensive.

the paper's main contributions can be summarized as follows:

1. **Addressing a Gap**: The paper tackles the challenge of training agents in modern video games, which has traditionally been resource-intensive and costly.
2. **Leveraging Large Vision Models**: It explores the potential of using publicly available large vision models to reduce costs and resource requirements, a pertinent issue given the current trend in machine learning towards larger models.
3. **Comparative Study**: A systematic study is conducted to compare the performance of publicly available visual encoders with traditional, task-specific, end-to-end training approaches in the context of imitation learning.
4. **Focus on Modern Games**: The study specifically targets modern video games, including Minecraft, Minecraft Dungeons, and Counter-Strike: Global Offensive, reflecting a move beyond simpler, classic game environments.
5. **Human-like Gameplay**: The authors emphasize training agents to play games in a human-like manner, using behavior cloning and offline training with human gameplay data, which is a step towards creating AI that can interact in complex environments in a natural way.

**Strengths:**

1. The authors have selected a diverse set of modern video games, including Minecraft, Minecraft Dungeons, and Counter-Strike: Global Offensive, for their experimental studies. This choice reflects a significant step forward from the commonly used Atari games in previous research, providing a more realistic and challenging benchmark for evaluating imitation learning techniques.
2. The paper introduces an innovative approach to imitation learning by leveraging publicly available large vision models. This strategy not only addresses the resource-intensive nature of training agents in modern video games but also democratizes access to high-quality training for smaller research groups or institutions.
3. The writing is clear, concise, and well-structured.

**Weaknesses:**

1. It seems that the task is very simple, such as chopping trees in Minecraft, which is a fairly straightforward task. The author's conclusion is that there is no significant difference between various visual encoders and input image resolutions. However, due to the simplicity of the task, this conclusion is unreliable. Evaluating models like CLIP and DINO on such simple tasks does not effectively demonstrate the differences between modern vision transformers and CNNs. I strongly recommend that the author choose more challenging tasks, such as `MineRLObtainDiamondShovel-v0` or `MineRLBasaltBuildVillageHouse-v0` etc.
2. In time-series decision-making tasks, the memory of historical states is crucial for making decisions. For example, VPT uses a transformer to record a history state of 128 frames, while the paper only utilizes LSTM to capture a limited number of frames. This can have negative implications for completing long-horizon tasks using behavior cloning.
3. Recently, the popular technique of Segment Anything has achieved better results in various visual tasks. The author can further compare this model to explore its potential.
4. In the conclusions shown in Table2, the best model of tree pruning only has a success rate of 32%, much lower than VPT's nearly 100%. Does this imply that the visual encoder is not actually the most important module in game playing?

In conclusion, although the author has compared a considerable number of vision encoders in the game, the reliability of the results is compromised due to their choices in task setting and temporal transformer.

Some relevant work has not been cited:
1. Open-world multi-task control through goal-aware representation learning and adaptive horizon prediction
2. A generalist agent
3. GROOT: Learning to Follow Instructions by Watching Gameplay Videos
4. Learning to drive by watching youtube videos: Action-conditioned contrastive policy pretraining

**Questions:**

See in weakness.

**Details Of Ethics Concerns:**

Null

---

> ### Author Response · Authors · 2023-11-16
> **Rebuttal to Reviewer yJPq (1/2)**
>
> We thank the reviewer for their comments and suggestions. Below, we will address the main comments raised by the reviewer.
>
> > It seems that the task is very simple, such as chopping trees in Minecraft, which is a fairly straightforward task. The author's conclusion is that there is no significant difference between various visual encoders and input image resolutions. However, due to the simplicity of the task, this conclusion is unreliable. Evaluating models like CLIP and DINO on such simple tasks does not effectively demonstrate the differences between modern vision transformers and CNNs. I strongly recommend that the author choose more challenging tasks, such as MineRLObtainDiamondShovel-v0 or MineRLBasaltBuildVillageHouse-v0 etc.
>
> We acknowledge the simplicity of the Treechop task in Minecraft but would like to highlight that 1. our goal is to cover a range of visually and conceptually diverse modern video games (breadth) rather than focusing on few challenging tasks in a single game (depth), and that 2. Minecraft Dungeons in particular represents a highly challenging decision making task. For a more detailed discussion of this topic, we refer the reviewer to the section “Evaluation Tasks” in our common response.
>
> > In time-series decision-making tasks, the memory of historical states is crucial for making decisions. For example, VPT uses a transformer to record a history state of 128 frames, while the paper only utilizes LSTM to capture a limited number of frames. This can have negative implications for completing long-horizon tasks using behavior cloning.
>
> We agree with the reviewer that memory of historical states can be crucial for decision making in some long-horizon tasks. However, we would like to disentangle the episodic length of tasks and the memory agents need to solve a task. Some long-horizon tasks require very limited or no memory to solve. For example, DQN was able to achieve superhuman performance in many Atari games without any memory, some of which feature relatively long episodes [1].
>
> Furthermore, we disagree that LSTMs are only able to capture a limited number of frames since they can learn to keep information in its hidden state (memory) throughout an episode as long as it provides value. Impressive prior decision making achievements in long-horizon tasks have been achieved using LSTM models [2]. In contrast, a transformer is always limited to the context of its input frames (dependent on the used context length)
>
>
> To further stress the relevance of agents maintaining history in our evaluation tasks, we highlight that the Minecraft Dungeons task requires agents to take up to 3,000 actions within a single episode (5mins at 10Hz). We believe that the ability of agents to effectively act in such a long-horizon task demonstrates that agents with LSTMs are sufficient to complete many long-horizon tasks in modern video games.
>
> > Recently, the popular technique of Segment Anything has achieved better results in various visual tasks. The author can further compare this model to explore its potential.
>
> We agree that Segment Anything appears to be a promising pre-trained visual model. However, its inference cost makes it infeasible for our tasks in which agents have to take actions in real-time. We refer the reviewer to the section “Segment Anything Pre-Trained Visual Encoder” of our common response for more details.
>
> > In the conclusions shown in Table2, the best model of tree pruning only has a success rate of 32%, much lower than VPT's nearly 100%. Does this imply that the visual encoder is not actually the most important module in game playing?
>
> We refer the reader to the section “Discussion of Related Work” in our common response where we contrast VPT to our evaluation setup in more detail.
>
> Furthermore, we politely but strongly disagree that our best models performing notably worse than VPT implies that the visual encoder is not the most important module in game playing. There are many differences between our evaluation setup and VPT (the most notable ones being the scale of data and models, as well as agent architecture) which all confound the comparison, so no such conclusions can be drawn from the discrepancy in results in our paper and VPT.
>
> > Some relevant work has not been cited
>
> We thank the reviewer for pointing out these recent, concurrent related works and included these works in Section 2 on related work. We further refer the reviewer to the section “Discussion of Related Work” in our common response where we discuss the work on open-world multi-task control in more detail. Lastly, we would like to point out that the most recent version of 1. and the listed paper 3. were made publicly available on arXiv after the ICLR submission deadline.

---

> > ### Author Response · Authors · 2023-11-16
> > **Rebuttal to Reviewer yJPq (2/2)**
> >
> > ## Citations
> > [1] Mnih, Volodymyr, Koray Kavukcuoglu, David Silver, Andrei A. Rusu, Joel Veness, Marc G. Bellemare, Alex Graves et al. "Human-level control through deep reinforcement learning." nature 518, no. 7540 (2015): 529-533.
> >
> > [2] Vinyals, Oriol, Igor Babuschkin, Wojciech M. Czarnecki, Michaël Mathieu, Andrew Dudzik, Junyoung Chung, David H. Choi et al. "Grandmaster level in StarCraft II using multi-agent reinforcement learning." Nature 575, no. 7782 (2019): 350-354.

---

> > > ### Comment · Reviewer_yJPq · 2023-11-17
> > > **Comments on Rebuttal**
> > >
> > > 1. The author's response did not solve the core issue we raised, which is that the task chosen by the author is too simple.
> > > 2. The author achieved only a 4% success rate using Impala CNN, while VPT with the same structure can almost complete this task with a 100% success rate. Maybe the authors should check the code or settings to avoid possible errors.
> > > After considering the author's rebuttal, I have decided to keep the score.

---

> ### Author Response · Authors · 2023-11-17
> **Response to Rebuttal Comments of Reviewer yJPq**
>
> We thank the reviewer for their prompt response but, with respect, we firmly disagree with both of their criticisms for the following reasons.
>
> ## Environment Selection
>
> We notice that the reviewer’s criticisms solely focus on our task selection in Minecraft and would like to highlight that we also evaluate in CS:GO and Minecraft Dungeons - the latter representing an entirely new environment for imitation learning. **Minecraft Dungeons is not a variation or task in Minecraft but a completely different game with distinct game mechanics** (focus on navigation and combat) **and visual perspective** (isometric top-down view in contrast to first-person view in Minecraft and CS:GO).
>
> To visualise the uniqueness and complexity of Minecraft Dungeons as well as the Minecraft and CS:GO environments, we would like to point the reviewer to our **released rollout videos**: https://anonymous.4open.science/r/imitation_learning_in_modern_video_games_rollouts-4C48/README.md
>
> ## Comparison to VPT
>
> We strongly disagree with the reviewer’s comparisons to VPT and their statement that VPT has the “same structure”. We would argue that adding comparisons to VPT, which would only be possible in Minecraft since VPT has been specifically designed and trained for this environment, would provide no additional value for our study. Given the many differences between VPT and our models, VPT outperforming our models in Minecraft provides no clear insight in addition to our findings.
>
> To expand on our common response, we would like to highlight the following differences:
>
> - **>100x parameters**: VPT’s main model has 500M parameters vs our model having ~4M (in the policy) + ~94K (in the referenced Impala CNN encoder) parameters
> - **~100,000x training data**: VPT uses ~70,000h of (filtered) training data vs ~45min training data for our Minecraft experiments
> - **>3,000x training budget**: VPT training uses 6480 GPU days (720 GPUs for 9 days) vs the training of the Impala CNN model using 2 GPU days (a single GPU for less than 2 days)
>
> Even if we focus on the ResNet architecture of the VPT policy model alone, the main VPT model’s ResNet has 40M parameters (the smallest variation has 10M parameters in the ResNet). In comparison, the Impala CNN model highlighted by the reviewer has ~94K parameters. The ResNet encoders considered in our study, which are architecturally more comparable to VPT’s ResNet, only have ~580K parameters which is still substantially less than the ResNet used in VPT.
>
> **The insights of our study are not conflicting or in competition with VPT, which serves as a foundation model for decision making in Minecraft, but provide additional and orthogonal findings about the impact of visual encoders for decision making in complex video games more broadly.**
>
> ## Code Quality
>
> Given the significant differences between VPT and our Impala CNN, we disagree with the assessment that the poor performance of the Impala CNN model suggests possible errors in our code. **We will open source the code to enable reproducibility** and assure the reviewer of the quality of the code.

---

### Author Response · Authors · 2023-11-16
**Common Rebuttal (1/3)**

# Common Rebuttal (1/3)

We thank all reviewers for their comments and thoughtful feedback. We are encouraged to hear that reviewers consistently found our writing clear, identified our studied problem as important to the community, and regarded our studied games as challenging benchmarks that reflect a significant step forward from the commonly used Atari games in previous research.

To further visualise the complexity of Minecraft Dungeons, we release a set of videos featuring rollouts of trained agents in the evaluation task. These videos can be accessed at the following anonymous repository: https://anonymous.4open.science/r/imitation_learning_in_modern_video_games_rollouts-4C48/README.md

We also revised the paper PDF to address comments of reviewers with all changes highlighted in orange colour.

Below, we address points made by several reviewers.

## Evaluation Tasks

Several reviewers point out the simplicity of the chosen Treechop task in Minecraft and suggest more challenging tasks within Minecraft.

First, we acknowledge the relative simplicity of the “tree chop” task, but also argue it is not trivial. The best BC-based solution in the MineRL 2021 competition reached 63% (vs. 32% in our work) log obtaining rate (see [1], “WinOrGoHome” team results), and this team optimised the training specifically for the “tree chop” task. This task is challenging because the agent has to be able to navigate the terrain effectively to reach (and recognize) a tree, and then attack a specific block of a tree for 3 in-game seconds straight (60 actions at 20Hz at which MineRL runs). This is especially difficult without task-specific tuning, as  a stochastic policy will occasionally sample a non-attack action and reset the chopping, or alternatively steer the mouse away from the block. We believe that studying behaviour broadly across Minecraft Dungeons, Minecraft Treechop, and CS:GO Aim Training is more informative for our study focused on the efficacy of visual encoders for imitation learning than studying a variety of complex tasks in Minecraft alone.

Second, as our motivation focuses on leveraging demonstration data from humans, we were constrained to tasks with available datasets of human demonstrations. For Minecraft, we filtered parts of the VPT contractor dataset to obtain clean examples of treechop. However, this type of data was not as readily available for other Minecraft tasks (e.g., there are no clean trajectories for obtaining diamonds in the VPT dataset). Previous MineRL datasets do contain such demonstrations, but only contain images at a lower resolution (64x64) which is not sufficient for studying our visual encoders.

Third, we would like to highlight the diversity of our evaluation tasks. Minecraft Dungeons, Minecraft, and CS:GO all feature different perspectives (first-person vs isometric perspective), graphical styles (highly stylistic visuals vs more realistic visuals), and/ or concepts (zombies, skeletons, swords, guns, etc.) which visual encoders need to identify and represent to enable effective decision making.  Given our motivation to study visual encoders for decision making, we believe that this diversity is important to cover a breadth of common scenarios in modern video games.

Lastly, we notice that reviewers largely focus on our evaluation and task selection in Minecraft and would like to highlight the complexity of the Minecraft Dungeons evaluation task. For this task, the agent needs to take up to 3,000 actions (5 minutes at 10Hz), which requires long contextual awareness in a vast action space featuring four continuous actions as well as eleven discrete actions. The agent needs to navigate across a complex, long, and visually diverse environment and combat enemies at stochastic locations and configurations. This demonstrates the ability of our trained agents to tackle challenging tasks which require (1) long-horizon decision making, (2) adaptation to stochastic environments, and (3) complex action selection across a comparably large action space.

To visualise the complexity of the considered tasks, in particular Minecraft Dungeons, we release a set of videos featuring rollouts of trained agents in the evaluation task at the following anonymous repository: https://anonymous.4open.science/r/imitation_learning_in_modern_video_games_rollouts-4C48/README.md

---

> ### Author Response · Authors · 2023-11-16
> **Common Rebuttal (2/3)**
>
> ## Discussion of Related Work
>
> Several reviewers mentioned Video Pretraining (VPT) [2] and work on Open-World Multi-Task Control [3] which we would like to further discuss.
>
> First, reviewers ask why VPT reaches higher performance than our trained models in Minecraft and suggest including VPT as a baseline. We would like to highlight that VPT trains a transformer model with 500M parameters on 70,000 hours of filtered IDM-labelled data using 720 V100 GPUs for 9 days. In contrast, we train LSTM models with ~4M parameters (excluding the visual encoder) on less than 1h of filtered data using a single A6000 GPU for a few hours.
>
> Given the vast discrepancy in used data and model sizes, we believe that any comparison to VPT would not be fair since VPT is clearly expected to perform notably better. To identify the contribution of visual encoders to the decision making performance in comparison to the results of VPT, we would need to compare visual encoders at a comparable scale of model size and training data. However, such comparison is infeasible due to its computational cost.
>
> Lastly, we would like to highlight that VPT focuses on Minecraft as an open-ended decision making benchmark, training only a few models at large scale. Instead, our study aims to demonstrate wider breadth by evaluating a total of 22 different visual encoder configurations across three modern video games.
>
> Regarding the work on open-world multi-task control [3], we thank the reviewers for pointing us to this related work which we now include in Section 2. This work leverages the VPT agent to collect demonstration data in Minecraft. First, we agree that this approach of leveraging pre-trained models is interesting. However, we would like to highlight that it relies on released pre-trained agent models which are not available in most video games. In this case, the designed approach would only be applicable in Minecraft since no similar pre-trained agent is available in Minecraft Dungeons or CS:GO. Secondly, we would like to point out that the dataset of this work was only released three weeks prior to the ICLR deadline (as per Github commits) which did not allow us to consider the collected data in our experiments. Thirdly, the work points out the difficulty of multiple tasks in open-ended environments, such as Minecraft, being indistinguishable from each other. In light of this challenge, we believe that the selection of effective visual encoders can be complementary to the proposed approach, since these have to perceive and represent task-relevant and discriminative information across different scenes.
>
>
> ## Fine-tuning of Pre-Trained Visual Encoders
>
> Reviewers sq4s and bAy4 suggest to additionally fine-tune the considered pre-trained visual encoders in our evaluation study. We agree that fine-tuning these visual encoders would add valuable comparisons to the evaluation study. However, fine-tuning these models with, in some cases, hundreds of millions of parameters is computationally very costly.
>
> To illustrate the computational cost, we would like to highlight that fine-tuning these visual encoders would not just require notably more compute (in particular VRAM) but also dramatically slow-down training. As outlined in Appendix G, our current experiments with pre-trained visual encoders first compute the embeddings with these frozen visual encoders for our entire dataset before training a policy network on these pre-computed embeddings. This process significantly speeds-up training (we found training to be 10-20x faster on pre-computed embeddings). Fine-tuning would prevent us from pre-computing embeddings for training since embeddings would change as the visual encoders are updated. At such computational cost, an empirical study of our scale (training 10 pre-trained visual encoders across three domains, each with three seeds) would not have been feasible.
>
> Lastly, we would like to highlight that our work to study the efficacy of varying visual encoders (pre-trained and end-to-end trained) for imitation learning in modern video games is largely motivated by the prohibitive cost of training decision making agents in these domains. We believe that our findings, even in the absence of fine-tuned pre-trained encoders, provides significant value to the community and can serve as a starting point for anyone to develop capable decision making agents with limited resources.

---

> ### Author Response · Authors · 2023-11-16
> **Common Rebuttal (3/3)**
>
> ## Segment Anything Pre-Trained Visual Encoder
>
> Reviewers yJPq and JgYZ suggested to additionally consider the pre-trained Segment Anything (SAM) [4] models. We agree with both reviewers that these models have demonstrated impressive generalisation capabilities. We considered including SAM pre-trained visual encoders in our study, but found these models to be too slow at inference time. On hardware available to us (including A6000, GTX 1080Ti, and V100 GPUs), computing embeddings with even the smallest SAM checkpoint took between 0.1 and 0.5 seconds which is too slow for decision making in video games. In CS:GO and Minecraft Dungeons, we aim to take actions at 16Hz and 10Hz, respectively, in real-time. This was not possible with pre-trained SAM models.
>
> ## Hyperparameter Selection
>
> Several reviewers asked how we selected hyperparameters for our experiments. Hyperparameters such as the learning rate, batch size, and model architecture configurations were determined in prior experiments in Minecraft Dungeons. To keep comparisons fair and focused on the visual encoders, we kept hyperparameters consistent in following experiments. We only reduced the total training duration in tasks which required notably less training (as seen in training loss curves in Appendix B) to limit the computational cost of experiments.
>
> We prioritised minimising conflating variations besides the variation of visual encoders to draw informative conclusions from our experiments. Furthermore, to conduct an extensive hyperparameter search for each considered configuration of visual encoders and tasks would be infeasible for our study where we consider a total of 22 different visual encoder configurations across three tasks.
>
> ## Revised Paper
>
> Lastly, we would like to highlight several improvements made in the paper based on the reviewers’ feedback. For details, we refer to the revised PDF which includes all changes highlighted in orange colour. Below, we list the most notable changes:
> - Prompted by our results in CS:GO, we hypothesised that the poor performance of DINOv2 could be explained by the image processing necessitated by the CS:GO dataset providing images at 280x150 resolution. Following this hypothesis and comments by reviewer bAy4, we now provide an evaluation of four pre-trained visual encoders in Minecraft (Appendix E), comparing the original online performance with the performance observed when alternating the image processing in two ways. We find that pre-trained visual encoders in Minecraft are more robust to alternative image processing as hypothesised from our CS:GO results, and incorporate these findings in Section 5.5.
> - Appendix B now includes loss curves for the training of all models in all three games (the paper previously only included the training loss in Minecraft Dungeons).
> - Following the suggestion of reviewer JgYZ, Appendix A now includes details on the image augmentation considered when training end-to-end visual encoders.
> - Following the suggestion of reviewer bAy4, Sections 5.3 and 5.4 now include further conclusions on the correlation between the number of parameters of pre-trained visual encoders and the online evaluation performance. In short, we observe no clear correlation between the size of pre-trained visual encoders and the resulting online performance, suggesting that larger pre-trained visual encoders might not always be beneficial for decision making in video games.
> - Following references provided by reviewers yJPq, JgYZ and bAy4, we incorporated recent work into Section 2 on related work.
>
> ## Citations
>
> [1] Kanervisto, Anssi, Stephanie Milani, Karolis Ramanauskas, Nicholay Topin, Zichuan Lin, Junyou Li, Jianing Shi et al. "Minerl diamond 2021 competition: Overview, results, and lessons learned." NeurIPS 2021 Competitions and Demonstrations Track (2022): 13-28.
>
> [2] Baker, Bowen, Ilge Akkaya, Peter Zhokov, Joost Huizinga, Jie Tang, Adrien Ecoffet, Brandon Houghton, Raul Sampedro, and Jeff Clune. "Video pretraining (vpt): Learning to act by watching unlabeled online videos." Advances in Neural Information Processing Systems 35 (2022): 24639-24654.
>
> [3] Cai, Shaofei, Zihao Wang, Xiaojian Ma, Anji Liu, and Yitao Liang. "Open-world multi-task control through goal-aware representation learning and adaptive horizon prediction." In Proceedings of the IEEE/CVF Conference on Computer Vision and Pattern Recognition, pp. 13734-13744. 2023.
>
> [4] Kirillov, Alexander, Eric Mintun, Nikhila Ravi, Hanzi Mao, Chloe Rolland, Laura Gustafson, Tete Xiao et al. "Segment anything." arXiv preprint arXiv:2304.02643 (2023).

---

### Meta-Review · Area_Chair_wuq1 · 2023-12-09

**Metareview:**

This paper presents a study of various modern video encoders for imitation learning in video games. The paper is well written and  conducts a systematic study that compares different pre-trained visual encoders and from-scratch trained visual encoders in three video games. The reviewers agree that the paper is well written and easy to understand. However, the reviewers all find the experimental study quite limited and have asked for substantially more experiments given that there is little technical novelty.

**Justification For Why Not Higher Score:**

No technical novelty. The experiments are conducted on very simple minecraft tasks.

**Justification For Why Not Lower Score:**

N/A

---

### Decision · Program_Chairs · 2024-01-16

Reject